# Endoplasmic reticulum stress-independent activation of unfolded protein response kinases by a small molecule ATP-mimic

Aaron S Mendez[1,2†], Jennifer Alfaro[3†], Marisol A Morales-Soto[3,4†], Arvin C Dar[1,5], Emma McCullagh[3], Katja Gotthardt[2,6], Han Li[2], Diego Acosta-Alvear[2], Carmela Sidrauski[2], Alexei V Korennykh[2,7], Sebastian Bernales[3], Kevan M Shokat[1*], Peter Walter[2*]

[1]Department of Cellular and Molecular Pharmacology, Howard Hughes Medical Institute, University of California, San Francisco, San Francisco, United States; [2]Department of Biochemistry and Biophysics, Howard Hughes Medical Institute, University of California, San Francisco, San Francisco, United States; [3]Fundación Ciencia & Vida, Santiago, Chile; [4]Departamento de Ciencias Biológicas, Facultad de Ciencias Biológicas, Universidad Andrés Bello, Santiago, Chile; [5]Department of Structural and Chemical Biology, Mount Sinai Hospital, New York, United States; [6]Department of Chemical Biology, Max Planck Institute of Molecular Physiology, Dortmund, Germany; [7]Department of Molecular Biology, Princeton University, Princeton, United States

*For correspondence: Kevan. Shokat@ucsf.edu (KMS); peter@ walterlab.ucsf.edu (PW)

†These authors contributed equally to this work

**Abstract** Two ER membrane-resident transmembrane kinases, IRE1 and PERK, function as stress sensors in the unfolded protein response. IRE1 also has an endoribonuclease activity, which initiates a non-conventional mRNA splicing reaction, while PERK phosphorylates eIF2α. We engineered a potent small molecule, IPA, that binds to IRE1's ATP-binding pocket and predisposes the kinase domain to oligomerization, activating its RNase. IPA also inhibits PERK but, paradoxically, activates it at low concentrations, resulting in a bell-shaped activation profile. We reconstituted IPA-activation of PERK-mediated eIF2α phosphorylation from purified components. We estimate that under conditions of maximal activation less than 15% of PERK molecules in the reaction are occupied by IPA. We propose that IPA binding biases the PERK kinase towards its active conformation, which *trans*-activates apo-PERK molecules. The mechanism by which partial occupancy with an inhibitor can activate kinases may be wide-spread and carries major implications for design and therapeutic application of kinase inhibitors.

## Introduction

Roughly 30% of all proteins encoded in eukaryotes pass through the endoplasmic reticulum (ER), where they are folded, modified, and assembled, before they are delivered to the plasma membrane, the outside of the cell, or to various way-stations in the secretory and endocytic pathways. To ascertain fidelity during protein maturation, ER-resident unfolded protein sensors continuously monitor the folding status in the ER lumen. The unfolded protein response (UPR) is induced when the protein folding capacity of the ER is surpassed, triggering the activation of three transmembrane sensors/signal transducers, IRE1 (inositol-requiring enzyme 1), PERK (protein kinase RNA (PKR)-like ER kinase), and ATF6 (activating transcription factor-6). Two of the sensors, IRE1 and PERK, are protein kinases that are amenable to modulation by small molecule ATP-mimetics.

**eLife digest** Cells contain thousands of proteins that carry out the essential tasks needed for survival. Before they can work, proteins must first fold into specific three-dimensional shapes. The endoplasmic reticulum, a cellular compartment that specializes in properly folding newly made proteins into their native states, is critical for this protein maturation process. If folding-enzymes in the endoplasmic reticulum are not properly balanced with the load of proteins they must fold, the endoplasmic reticulum can be overwhelmed with unfolded proteins that accumulate, leading to 'endoplasmic reticulum stress'.

The cell copes with endoplasmic reticulum stress by triggering the 'unfolded protein response' (UPR). This response helps to clear the unfolded proteins by increasing the size of the endoplasmic reticulum and the concentration of folding enzymes within it, and by decreasing the influx of newly made protein into the endoplasmic reticulum. The UPR engages signaling molecules in the endoplasmic reticulum membrane, among them two signaling enzymes called IRE1 and PERK. Drugs that activate these signaling enzymes could help the cell to deal with unfolded proteins, prevent toxicity resulting from endoplasmic reticulum stress, and ward off the diseases that result from it.

Mendez, Alfaro, Morales-Soto et al. developed a small molecule, called IPA (short for IRE1/PERK Activator), that was designed to bind to and activate IRE1. Serendipitously, IPA not only activated IRE1 but also activated PERK. Surprisingly, PERK activation was only observed at low IPA concentrations in which IPA occupied the active sites in only a few PERK molecules, whereas at higher concentrations and full occupancy IPA completely inhibited PERK. Mendez, Alfaro, Morales-Soto et al. proposed that, under conditions of partial IPA occupancy, a minority of IPA-bound PERK molecules assume an activated state that propagates to adjacent PERK molecules that have no IPA bound to them, and activates them.

Similar dose-dependent activation was previously observed for a clinically used drug designed to inhibit a similar signaling enzyme that is important in cancer progression. Together with the observations of Mendez, Alfaro, Morales-Soto et al., these results suggest that research into similar treatments must consider that a 'minimal dose' can exist, below which drugs may have the opposite effect to what is desired. Further work is still needed to fully understand the mechanisms that produce such behavior.

IRE1is the most conserved of these proteins. It contains an ER-lumenal sensor domain that is activated by binding directly to unfolded polypeptides (*Credle et al., 2005*; *Gardner and Walter, 2011*). As a result, IRE1 oligomerizes, activating its cytosolic kinase and endoribonuclease domains (*Cox et al., 1993*; *Sidrauski et al., 1996*; *Calfon et al., 2002*; *Korennykh et al., 2009*; *Li et al., 2010*). IRE1's RNase domain initiates a non-conventional splicing reaction that results in the excision of an intron from the mRNA encoding the transcription factor XBP1. XBP1 produced from the spliced mRNA drives transcription of UPR target genes to remedy ER stress. The luminal domain of PERK is homologous to that of IRE1 and thus its activation is presumably also driven by direct binding to unfolded polypetides (*Gardner and Walter, 2011*). Active PERK phosphorylates the α-subunit of eukaryotic translation initiation factor 2 (eIF2α) (*Harding et al., 1999*), leading to trapping eIF2α in its GDP-bound inactive state, blocks eIF2α recycling. As a result, global protein synthesis is attenuated, while a few mRNAs, including that encoding the transcription factor ATF4, are preferentially translated (*Harding et al., 2000*; *Wek et al., 2006*).

Recent chemical genetic work in our laboratories revealed that phospho-transfer by Ire1's kinase domain can be bypassed using an ATP mimetic (1NM-PP1) (*Papa et al., 2003*; *Rubio et al., 2011*). Starting with studies in *Saccharomyces cerevisiae*, we showed that Ire1's RNase modality can be activated using a small molecule ATP mimetic (1NM-PP1), when used in conjunction with a mutant form of Ire1 ('IRE1-as' for analog sensitized) that allows 1NM-PP1 to bind to the ATP-binding site of IRE1's kinase domain. Subsequent work showed that the RNase activity of wild-type Ire1 can too be activated pharmacologically with the broad-acting kinase inhibitors APY29 and Sunitinib in vitro (*Korennykh et al., 2009*). The crystal structure of the Ire1 kinase/RNase domains bound to APY29, combined with biophysical and enzyme kinetic analyses, showed that binding of ATP-mimetic ligands to Ire1's active kinase site predisposes the enzyme to oligomerization, which activates its RNase activity.

Ligand binding to the ATP binding pocket in IRE1's kinase domain, however, does not always result in oligomerization and RNase activation. Rather, activation requires that IRE1's kinase domain is in its *active* conformational state, characterized by the inward positioning of the αC helix and the DFG-loop in the kinase active site (DFG/αC-in conformation) (*Korennykh et al., 2011*; *Korennykh and Walter, 2012*; *Wang et al., 2012*; *Sanches et al., 2014*). Thus, ATP-mimetic ligands that trap IRE1's kinase domain in the inactive, DFG/αC-out conformation act as inhibitors, rather than activators, of IRE1 oligomerization and signaling via its RNase domain. Because RNase activation can occur in the absence of a phospho-transfer reaction, IRE1 is unique in that it is possible to monitor the functional consequences of conformational changes in the kinase domain induced by ligand occupancy of the ATP-binding site without concerns of losing the kinase activity.

The model depicting IRE1's kinase domain as a switch that becomes trapped in two states (DFG/αC-in and DFG/αC-out) depending on the ligand bound to its active site is an over-simplification. Different ligands yield different plateaus of maximal oligomerization and RNase activation, even when saturating the active site. This seemingly perplexing property is reconciled by the model in which different ligands predispose IRE1's kinase domain to populate the DFG/αC-in and DFG/αC-out states to different degrees; a strong IRE1 RNase activator would stabilize the DFG/αC-in state, whereas a weaker one would bias the IRE1 molecules in the population towards the DGF/αC-in state, without completely trapping them in this state. The reverse would be true for IRE1 RNase inhibitors, which would bias IRE1's kinase domain towards the DFG/αC-out state.

To date, models of IRE1 activation have largely been derived from in vitro characterizations that lack in vivo confirmation, as the available tools were non-selective (and hence overtly toxic) to test in living cells (*Wang et al., 2012*). Moreover, while 1NM-PP1 predisposes IRE1-as towards activation, it proved insufficient to activate IRE1 in cells in the absence of ER stress (which vastly concentrates IRE1 by virtue of oligomerization of the lumenal domain) or over-expression. Here, we describe the development of a novel small molecule, IPA, as the lead compound of a series of second-generation IRE1 activators. Surprisingly, IPA activates not only IRE1's RNase, but also PERK signaling but, by contrast to its ability to activate IRE1, only at low concentrations. We propose that PERK activation results from ligand-induced conformational changes in a small percentage of the molecules in the population that then interact with and activate PERK molecules that contain an empty active site.

## Results

### Generation of small molecule activators of IRE1α

Recent work identified an ATP mimetic that activates mammalian IRE1α's RNase activity in vitro (*Wang et al., 2012*; *Sanches et al., 2014*). These results, along with the co-crystal structure of *S. cerevisiae* Ire1 with the aminopyrazole-based inhibitor APY29 (PDB ID: 3FBV) (*Korennykh et al., 2009*), provided a starting point to develop more selective and more potent IRE1 activators. We reasoned that (1) the cyclopropyl substituent on the pyrazole ring, which binds to the gatekeeper pocket in the *S. cerevisiae* Ire1 structure, would be a key determinant of human IRE1α binding, (2) interactions of the hinge-binding element of the APY29 scaffold would be essential to stabilizing IRE1α's kinase domain in a conformation leading to RNase activation, and (3) the pyrimidine ring, which occupies the adenine pocket in the *S. cerevisiae* structure, would provide an appropriate space filling moiety that further enhances affinity to the ATP binding pocket (*Figure 1A*). We therefore kept these three elements constant in further optimizations and explored varying substituents attached to the pyrimidine ring for their ability to improve properties of the compounds.

To this end, we modified the molecule by substituting urea-linked and variably *m*- and *p*-substituted phenyl groups for the benzimidazole of APY29. Diphenyl urea linkers have previously been used successfully in the design of kinase inhibitors (*Dar et al., 2008*; *Dar and Shokat, 2011*; *Korennykh et al., 2012*). In exploring the addition of this chemotype on the APY29 scaffold, we generated a panel of 10 variants (*Figure 1B*). Each member of the panel was tested in an RNase activity assay using recombinant IRE1α-KR43 and a labeled short hairpin RNA substrate derived from human *XBP1* mRNA. IRE1α-KR43 comprises the soluble cytoplasmic portion of IRE1α, composed of

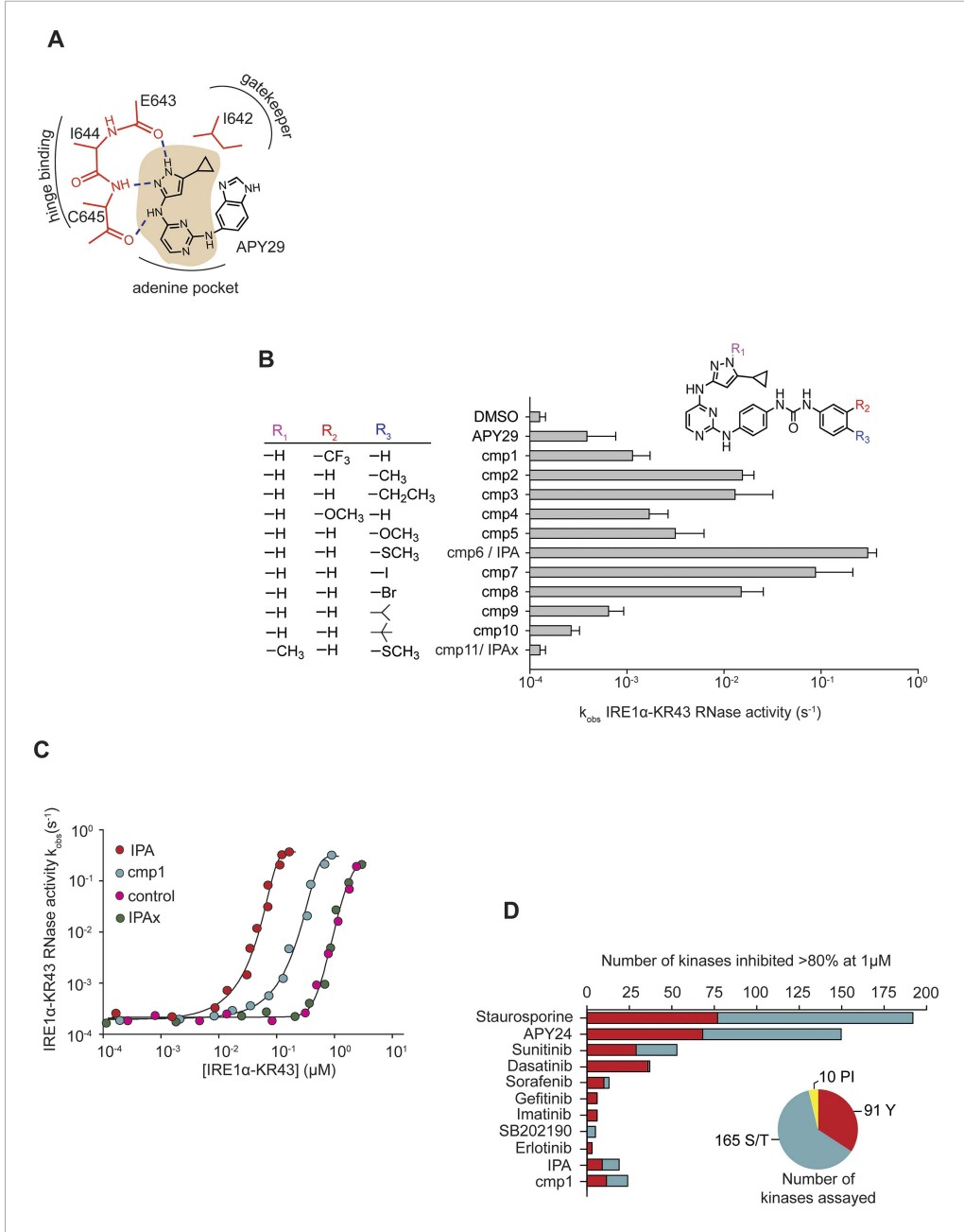

**Figure 1**. Design and characterization of IRE1α activators. (**A**) The core scaffold of APY29 (aminopyrazole pyrimidine-base indicated in beige). (**B**) Structure-activity analysis of activating compounds. Compounds were assayed at 1 µM in a RNA cleavage assay containing IRE1α-KR43 (200 nM) and 5′ [$^{32}$P]-labeled RNA substrate HP21 (see 'Materials and methods'). IPAx is methylated at the N[1] position in the pyrazole ring indicated as shown in R$_1$. (**C**) The effects of activating compounds (20 µM) as a function of IRE1α-KR43 concentration. $1/2k_{max, obs}$ and Hill coefficients ('n') were calculated (IPA: $1/2k_{max, obs}$ = 0.43 µM, n = 4.2; cmp1: $1/2k_{max, obs}$ = 3 µM, n = 3.8; IPAx $1/2k_{max, obs}$ = 8 µM, n = 3.3, DMSO control: $1/2k_{max, obs}$ = 8 µM, n = 3.3). (**D**) Compounds and FDA-approved drugs (all at 1 µM) were screened against a panel of 266 recombinant human kinases ('S/T': Ser/Thr protein kinases; 'Y': Tyr protein kinases, 'PI': phosphatidylinositide lipid kinases). APY24 contains the aminopyrozole pyrimidine-base, which is similar to APY29 (*Figure 1—figure supplement 2*).

The following figure supplements are available for figure 1:

**Figure supplement 1**. Concentration dependence of IPA, cmp1, and APY29 on IRE1α-KR43 RNase activity.

**Figure supplement 2**. Structure of APY based compound.

the kinase and RNase domains plus 43 amino acids of the linker that bridges the kinase to its transmembrane domain.

As expected from the chemical design strategy, we produced a series of IRE1α RNase activators. We observed that compounds bearing *m*-trifluoromethyl, *p*-methyl, *p*-ethyl, *m*-metoxy, and *p*-methoxy groups on the terminal phenyl ring (**cmp1**, **cmp2**, **cmp3**, **cmp4**, and **cmp5**) activated IRE1α-KR43 activity 10–100-fold under assay conditions, as compared to fivefold by the parent compound APY29 (*Figure 1B*). These results were surprising, because the presence of an *m*-trifluoromethyl group was previously shown to be important to retain activity of related kinase inhibitors but, as shown here, is not essential for activation of IRE1α-KR43. We were also surprised that the activity was improved further by the presence of a *p*-substituted thioether group and, similarly albeit weaker, by the presence of a *p*-iodo group (**cmp6** and **cmp7**, 900-fold RNase activation under assay conditions). We conclude that IRE1α prefers a substituent with polarizable character at the *para*-position of the terminal phenyl ring, perhaps indicating that the ring occupies a hydrophobic region in IRE1α's active site. Increasing bulkiness at the *para*-position resulted in decreased activities (**cmp8**, **cmp9**, and **cmp10**), suggesting that the size of this pocket must be limited.

We decided to further characterize compound **cmp6**, based on its robust activation properties. Henceforth, we refer to **cmp6** as IPA (for <u>I</u>RE1/<u>P</u>ERK <u>A</u>ctivator) for reasons to be discussed below. To validate that binding of IPA to IRE1α's kinase domain is critical for IRE1α RNase activation, we generated a control compound, IPAx. In IPAx, the pyrazole ring bears an additional N1-methyl group that is predicted to sterically interfere with binding to the gatekeeper pocket in IRE1α (*Figure 1A*). Indeed, when assayed for activation of IRE1α RNase activity, IPAx was inactive showing an indistinguishable effect from the DMSO control (*Figure 1B*).

ATP mimetics, such as APY29, induce IRE1α-KR43 oligomerization, which in turn activates the RNase activity (*Korennykh et al., 2011*; *Wang et al., 2012*). To assess whether IPA acts through a corresponding mechanism, we measured RNase activity as a function of enzyme concentration in the presence of saturating compound concentrations (20 µM; *Figure 1C* and *Figure 1—figure supplement 1*). When no activator was added, apo-IRE1α–KR43 RNase activity, measured as $k_{obs}$ under single-turnover conditions, increased sharply with increasing enzyme concentration, reaching ½ $k_{max, obs}$ at 8 µM (*Figure 1C*, purple circles). The measured Hill coefficient of the activation was n = 3.3, indicating that, consistent with previous work (*Li et al., 2010*), enzymatic activation parallels oligomerization. By contrast, in the presence of **cmp1** or IPA, IRE1α–KR43 RNase activity reached ½ $k_{max, obs}$ at 3- and 20-fold lower enzyme concentrations, respectively (½ $k_{max, obs}$ = 3 µM and 0.43 µM; and n = 3.8 and 4.2, *Figure 1C*, red and blue circles), indicating that **cmp1** and IPA binding significantly predispose IRE1α–KR43 to oligomerization and activation. By contrast, addition of IPAx did not enhance IRE1α activation, showing activation kinetics that were in all respects indistinguishable from the no-compound control (*Figure 1C*, pink circles), consistent with its inability to bind to IRE1α-KR43 (*Figure 1C* and *Figure 1—figure supplement 1*).

Kinome-wide screening of **cmp1** and IPA demonstrated a dramatic improvement of selectivity for both molecules when compared to APY24 (*Figure 1D* and *Figure 1—figure supplement 1*), a close analog of APY29, containing the same core-scaffold. In this assay, compounds were screened at a fixed concentration of 1 µM against a panel of 266 kinases, and the number of those inhibited by ≥ 80% was scored (*Figure 1D*). We also included in the analysis several bench-mark inhibitors, such as the promiscuous natural product staurosporine (STS) and several clinically approved kinase inhibitors. We note that both **cmp1** and IPA show better selectivity profiles when compared to the clinically approved compounds Sunitinib and Dasatinib.

## In vivo effects of IPA on IRE1α signaling

To examine the effects of IPA in living cells, we monitored the UPR in HEK293T cells using RT-PCR to measure splicing of *XBP1* mRNA. We found that IPA-induced *XBP1* mRNA splicing in a time- and dose-dependent manner (*Figure 2A*). At a concentration of 2 µM IPA, splicing was induced within 2 hr (lane 5). As expected, IPAx did not induce *XBP1* mRNA splicing (*Figure 2B*, lane 3). We used tunicamycin (Tm), which induces ER stress by blocking *N*-linked glycosylation, as a positive control to induce the UPR (*Figure 2A*, lanes 7, and *Figure 2B*, lane 4).

To further rule out off-target effects through which IPA might induce ER stress indirectly, we use a recently discovered IRE1α inhibitor, AD60, that binds to the ATP binding pocket of IRE1α's kinase

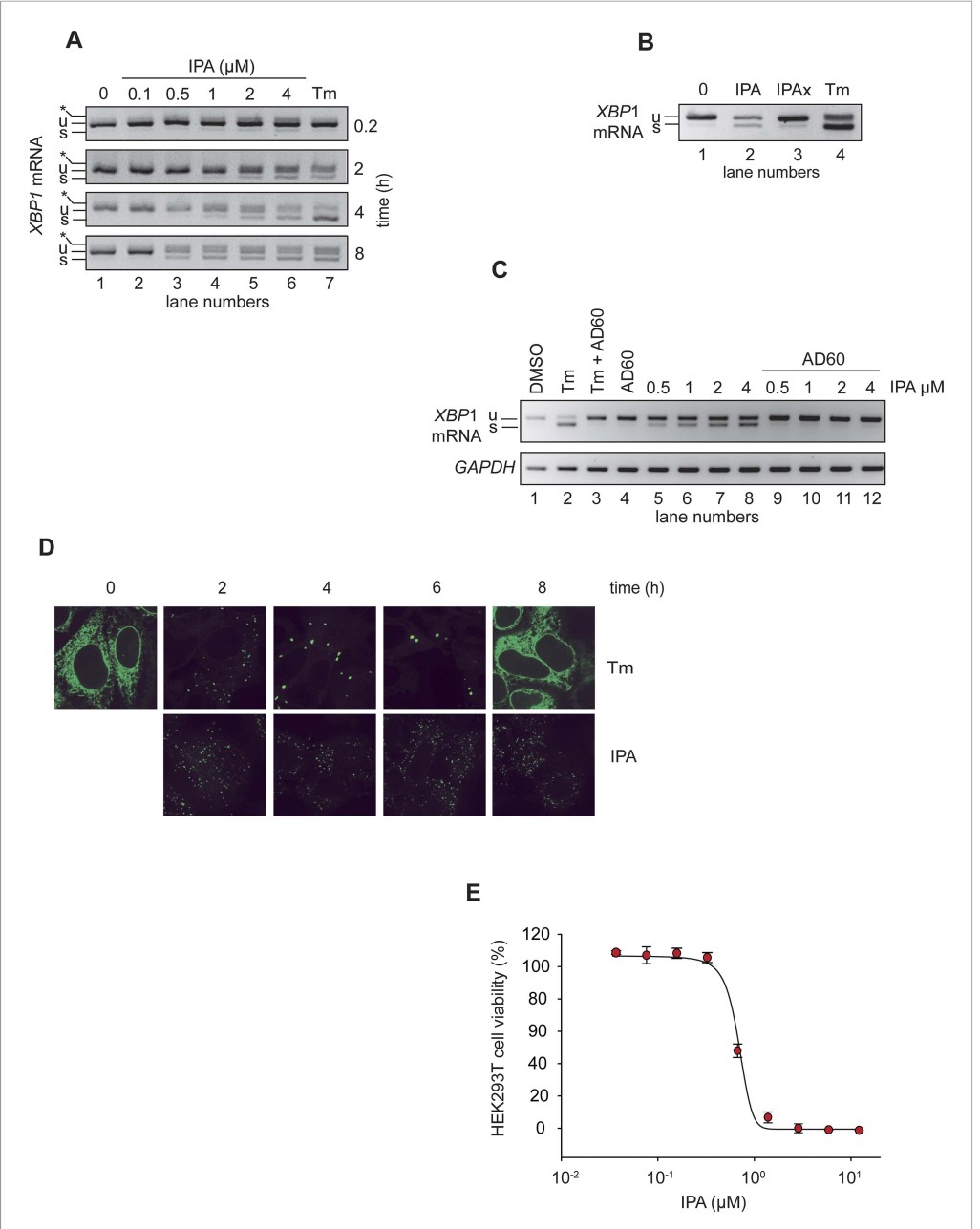

**Figure 2**. IPA activates the IRE1 branch of the unfolded protein response (UPR) in HEK293T cells. (**A**) HEK293T cells were treated with increasing concentrations of IPA as a function of time. Tunicamycin (Tm, 2 μg/ml) was used as positive control to induce endoplasmic reticulum stress. The resulting *XBP*1 mRNA spliced products detected by RT-PCR ('u': unspliced and 's': spliced) are indicated. Control cells were treated with DMSO only. The asterisk identifies a hybrid amplicon resulting from spliced and unspliced *XBP*1 mRNA. (**B**) The effects of IPAx on the splicing of *XBP*1 mRNA in HEK293T cells were detected by RT-PCR after 4-hr incubation ([IPA] and [IPAx] = 2 μM). (**C**) Inhibition of IPA-mediated *XBP*1 mRNA splicing in HEK293T cells by AD60 (incubation time = 4 hr; [AD60] = 1 μM). (**D**) IRE1-GFP foci formation in T-REx293 cells. IRE1-GFP was visualized by confocal microscopy. (**E**) Effects of IPA on HEK293T cell viability ($LD_{50}$ = 0.83 μM).

The following figure supplement is available for figure 2:

**Figure supplement 1**. AD60 inhibition of *XBP*1-luciferase-splicing reporter activation in HEK293T cells.

domain, locking it into its inactive DFG/αC-out conformation and inhibiting its RNase activity (**Dar et al., 2008**; **Korennykh et al., 2012**). Using an *XBP1* mRNA-splicing reporter that produces an *XBP1*-luciferase fusion protein only after IRE1-mediated splicing of its mRNA, we determined an $IC_{50}$ of 0.75 µM for AD60 in HEK293T cells (**Figure 2C** and **Figure 2—figure supplement 1**). As expected, AD60 inhibited Tm-induced *XBP1* mRNA splicing (**Figure 2C**, lane 3). Likewise, AD60 fully ablated IPA-induced *XBP1* mRNA splicing (**Figure 2C**, lanes 9–12), even at the highest IPA concentration tested. Taken together, these results confirm that IPA binds to the ATP-pocket of IRE1α to induce *XBP1* mRNA splicing in cells.

We next used a T-REx293 cell line, containing a genome-integrated doxycycline-inducible IRE1α-GFP fusion gene, to monitor IRE1α oligomerization in vivo (**Li et al., 2010**). In the presence of doxycycline, IRE1α-GFP was expressed and localized to the ER (**Figure 2D**). As previously described, Tm rapidly induced the relocalization of IRE1α-GFP into discrete foci, indicative of IRE1α oligomerization and activation (**Li et al., 2010**). Furthermore, the foci dissolved by 8 hr of treatment as cells attenuated IRE1α signaling even in the presence of unmitigated ER stress (**Figure 2D**, upper panels) (**Li et al., 2010**). As expected, IPA caused relocalization of IRE1α-GFP into foci, consistent with sustained IRE1 activation (**Figure 2D**, lower panels). While Tm-induced foci became larger and fewer over time as previously observed, IPA-induced foci remained small and numerous. Interestingly, IPA-induced foci remained stable even at late time points, consistent with the results obtained for *XBP1* mRNA splicing (**Figure 2A**). These results suggest that IPA-binding to IRE1α stabilizes IRE1α's oligomeric state, interfering with its attenuation.

We next tested the effect of IPA on cell viability. As shown in **Figure 2E**, IPA killed cells with an $LD_{50}$ of 0.82 µM. This strong toxicity was surprising considering that previous studies have shown that IRE1α has a cytoprotective role and that IRE1α knock-out cells are viable. The result therefore is best explained by a possible off-target effect connected to ER stress, likely to be mediated through another kinase in the cell. Moreover, as the interplay between the different UPR branches controls the switch over from cytoprotection to apoptosis (**Lin et al., 2007**), we next probed for possible effect of IPA on PERK and ATF6 signaling.

## In vivo effects of IPA on the ATF6 and PERK UPR branches

To assess the effects of IPA on the activity of the two other UPR branches, we first examined activation of ATF6. To this end, we monitored mRNA levels by RT-PCR for two ATF6-driven transcriptional target genes, *HERP1* and *DERL3*, in HEK293T cells. As expected, treatment with Tm induced the expression of these two genes (**Figure 3A,B**, lanes 6–9). By contrast, IPA caused no detectable induction (**Figure 3A,B**, lanes 2–5), indicating that IPA does not induce general ER stress.

To evaluate effects of IPA on the PERK branch, we measured the phosphorylation state of PERK and eIF2α by Western blotting. Upon activation, PERK becomes hyperphosphorylated and displays a characteristic mobility shift in SDS-PAGE gels, as observed in cells treated with Tm (**Figure 3C**, lane 7). Unexpectedly, we observed that IPA also induced a shift in the mobility of PERK (**Figure 3C**, lanes 2–4). This shift, however, was only observed at low-to-intermediate concentrations (0.1, 0.5, and 1 µM), whereas at higher concentrations (2 and 4 µM, **Figure 3C**, lanes 5 and 6), PERK phosphorylation was severely diminished or not detectable. We also monitored PERK activation by phosphorylation of eIF2α, detected with a phospho-eIF2α-specific antibody (**Figure 3C**). These results paralleled those obtained for PERK phosphorylation. By contrast, equivalent concentrations of IPAx did not stimulate PERK or eIF2α phosphorylation (**Figure 3D**, lane 3).

PERK activation by IPA could be due to direct binding of IPA to PERK, or it could be caused by indirect activation of PERK by IRE1. If the latter were true, IPA should not trigger PERK activation in the absence of IRE1. To test this notion, we treated mouse embryonic fibroblasts (MEFs) derived from *Ire1α$^{-/-}$* and *Perk$^{-/-}$* knock-out mice with IPA and monitored *XBP1* mRNA splicing, and PERK and eIF2α phosphorylation. As in HEK293T cells, treatment of wild-type MEFs with IPA induced both the PERK and IRE1 branches of the UPR. As expected, we observed no IPA-induced *XBP1* mRNA splicing in *Ire1α$^{-/-}$* MEFs (**Figure 3E**, lane 4) and only a trace amount of IPA-induced eIF2α phosphorylation in *Perk$^{-/-}$* MEFs (**Figure 3F**, lane 6). By contrast, we observed no IPA-induced *XBP1* mRNA splicing, but pronounced PERK phosphorylation (indicated by its mobility shift) and eIF2α phosphorylation in the *Ire1α$^{-/-}$* knock-out MEFs. These results indicate that IPA activates the IRE1 and PERK branches of the UPR independently.

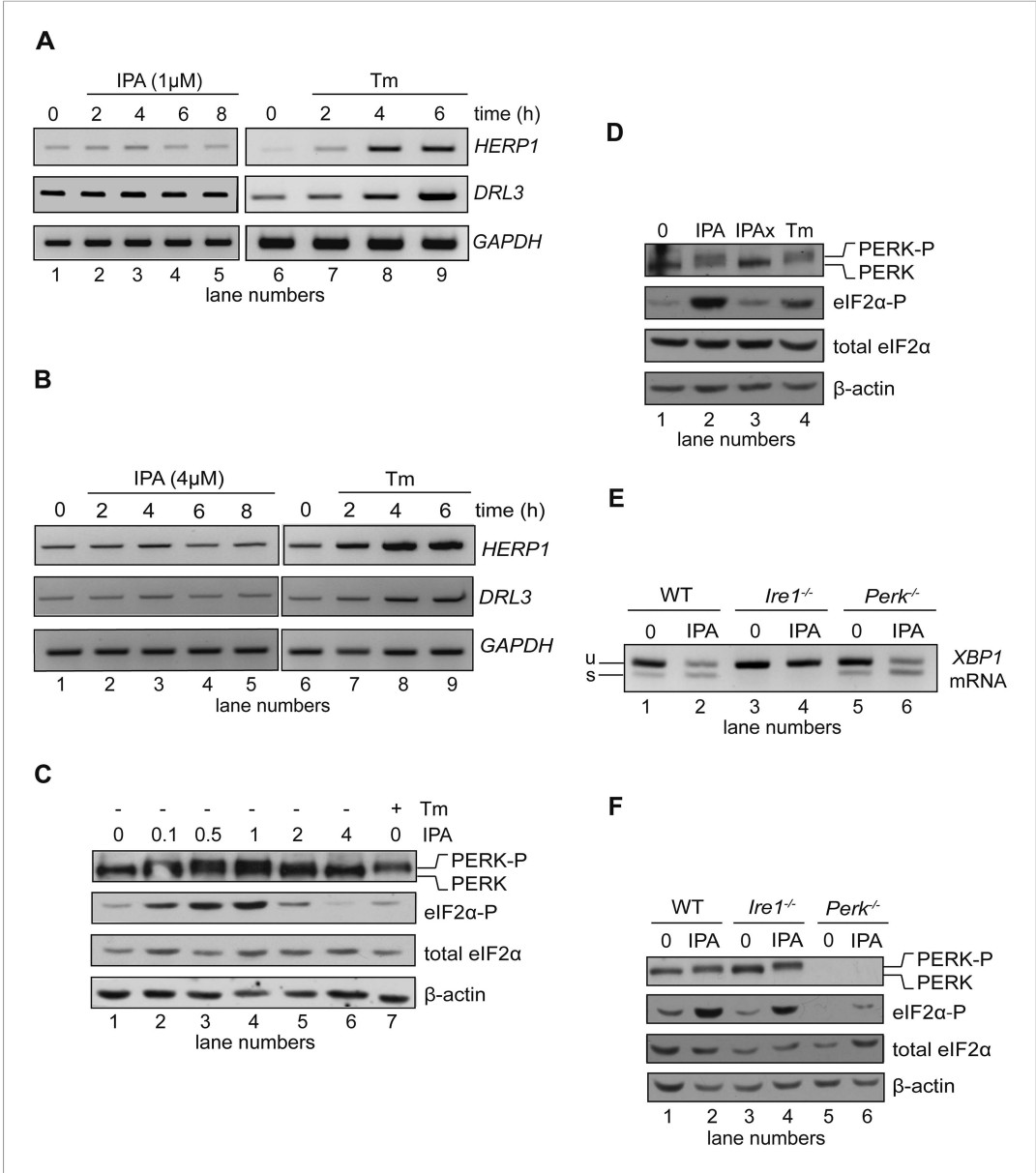

**Figure 3**. Effects of IPA on the ATF6 and PERK branches of the UPR. (**A**) The levels of ATF6 transcriptional target mRNAs *HERPUD*1 (*HERP1*) and *DERLIN*3 (DERL3) were measured by RT-PCR in HEK293T cells treated with 1 μM IPA. GAPDH was used as a loading control. (**B**) Same as in (**A**) but cells were treated with 4 μM IPA. (**C**) Phosphorylation of PERK and eIF2α in HEK293T cells treated with IPA were detected by immunoblotting (incubation time = 4 hr); PERK phosphorylation is apparent from its shift in gel mobility; eIF2α phosphorylation was detected using a phospho-specific antibody. β-actin was used as a loading control. (**D**) HEK293T cells were treated for 4 hr with IPA (1 μM), IPAx (1 μM), or Tm (2 μg/ml). PERK shift and eIF2α phosphorylation were detected as in **C**. (**E**) Wild-type (WT), *Ire1*[−/−], and *Perk*[−/−] mouse embryonic fibroblast (MEFs) cells were treated with IPA (1 μM) or Tm (5 μg/ml) for 4 hr. Induction of *XBP1* mRNA splicing by IPA was measured by RT-PCR. (**F**) Cells were treated as above. PERK gel mobility shift and eIF2α phosphorylation were detected by immunoblotting.

To test whether IPA binds PERK at its ATP binding site, we purified the cytosolic domain of PERK as a GST-PERK fusion protein. This preparation purified as a stable GST-PERK dimer, which in the presence of radiolabelled [γ-$^{32}$P]ATP efficiently phosphorylated eIF2α in vitro. We next tested the effects of IPA and IPAx using this kinase assay. IPA inhibited the PERK kinase reaction with an IC$_{50}$ = 2.8 μM (*Figure 4A*, solid red circles and *Figure 4—figure supplement 1*). These data suggest that IPA

binds to PERK directly, where it would act as a competitive inhibitor for ATP and thus block its phosphorylation activity.

## IPA activates PERK at low concentrations but inhibits it at higher concentrations

The observation of IPA-induced PERK *inhibition* (shown in *Figure 3C* and *Figure 4A*) poses the conundrum of why we observed IPA-induced *activation* in the low micromolar IPA range, producing an unusual, bell-shaped dose response (*Figure 3C*). This behavior could be explained by a model posing that at lower IPA concentrations, IPA occupies the ATP-binding sites of a subset of PERK molecules, triggering kinase activation of neighboring, unoccupied PERK molecules, as previously shown for other kinases (*Hall-Jackson et al., 1999*; *Hatzivassiliou et al., 2010*; *Poulikakos et al., 2010*). At higher IPA concentrations, IPA would saturate all PERK molecules, thereby inhibiting the pathway. We treated HEK293T with both Tm and IPA, expecting that Tm would not be able to activate PERK signaling in the presence of sufficiently high IPA concentration. Indeed, at 4 μM IPA, we observed that Tm addition did not activate PERK (*Figure 4B*, lane 6). Moreover, at 1 μM IPA, which induces PERK activation (*Figure 3C* and *Figure 3D*), we found that Tm did not enhance the PERK mobility shift or eIF2α phosphorylation further than what was achieved by IPA alone (*Figure 4B*, lane 5). IPA inhibition of PERK was reversible, since washing IPA out allowed for activation by Tm (*Figure 4B*, lane 9).

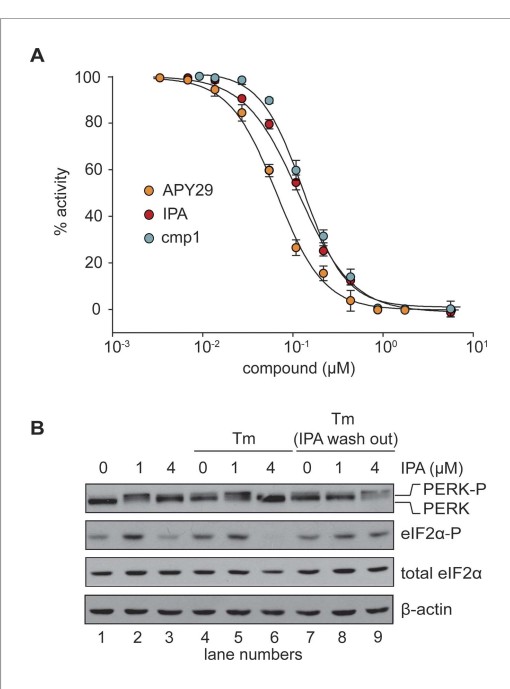

**Figure 4**. In vitro effects of IPA on GST-PERK kinase activity. (**A**) The effects of IPA, cmp1, and APY29 on GST-PERK kinase activity were monitored measuring phosphorylation of purified *S. cerevisiae* eIF2α with γ-[$^{32}$P]-labeled ATP (IPA IC$_{50}$ = 2.8 μM; cmp1 IC$_{50}$ = 4.5 μM, APY29 IC$_{50}$ = 0.69 μM). (**B**) HEK293T cells were pre-treated with IPA for 30 min. Cells were then either co-incubated with IPA and Tm (2 μg/ml) for an additional 4 hr (lanes 4, 5, and 6) or washed (IPA wash-out) and treated with Tm alone (lanes 7, 8, and 9).

The following figure supplement is available for figure 4:

**Figure supplement 1**. Audioradiographs of IPA, Cmp1 and APY29 tested against PERK-GST fusion protein.

## PERK inhibitors block IPA-induced PERK activation and restore translation

We next demonstrated that it is possible to keep the PERK pathway off in the presence of IPA. To this end, we added the selective PERK inhibitor GSK2606414 (*Axten et al., 2012*) (henceforth abbreviated as 'GSK') to HEK293T cells treated with IPA, at concentrations that exert the greatest effect on PERK pathway activation. As expected, GSK abolished IPA-induced PERK and eIF2α phosphorylation (*Figure 5A*; lanes 5 and 6 and *Figure 5—figure supplement 4*).

The phosphorylation of eIF2α resulting from PERK activation inhibits translation initiation. Indeed, when we incubated HEK293T cells with IPA in the low concentration range where PERK becomes activated, incorporation of [$^{35}$S]-methionine into newly synthesized proteins was impaired (*Figure 5B*, lanes 3 and 5), consistent with translation attenuation. Interestingly, treatment with GSK-restored translation to baseline levels (*Figure 5B*, lanes 4 and 6 and *Figure 5—figure supplement 1*). This pharmacological manipulation therefore allowed us to experimentally uncouple IPA-induced IRE1α and PERK activation. In particular, we were interested in testing whether the unexpected toxicity of IPA (*Figure 2E*) could, at last in part, be explained by PERK activation, which is known to induce apoptosis (*Lin et al., 2007*). To explore this notion, we tested cell viability at increasing IPA concentrations in the presence GSK. GSK inhibits PERK in cells with an EC$_{50}$ of 200 nM (*Axten et al., 2012*; *Moreno et al., 2013*) and shows no cell toxicity at 1 μM (*Figure 5C* and

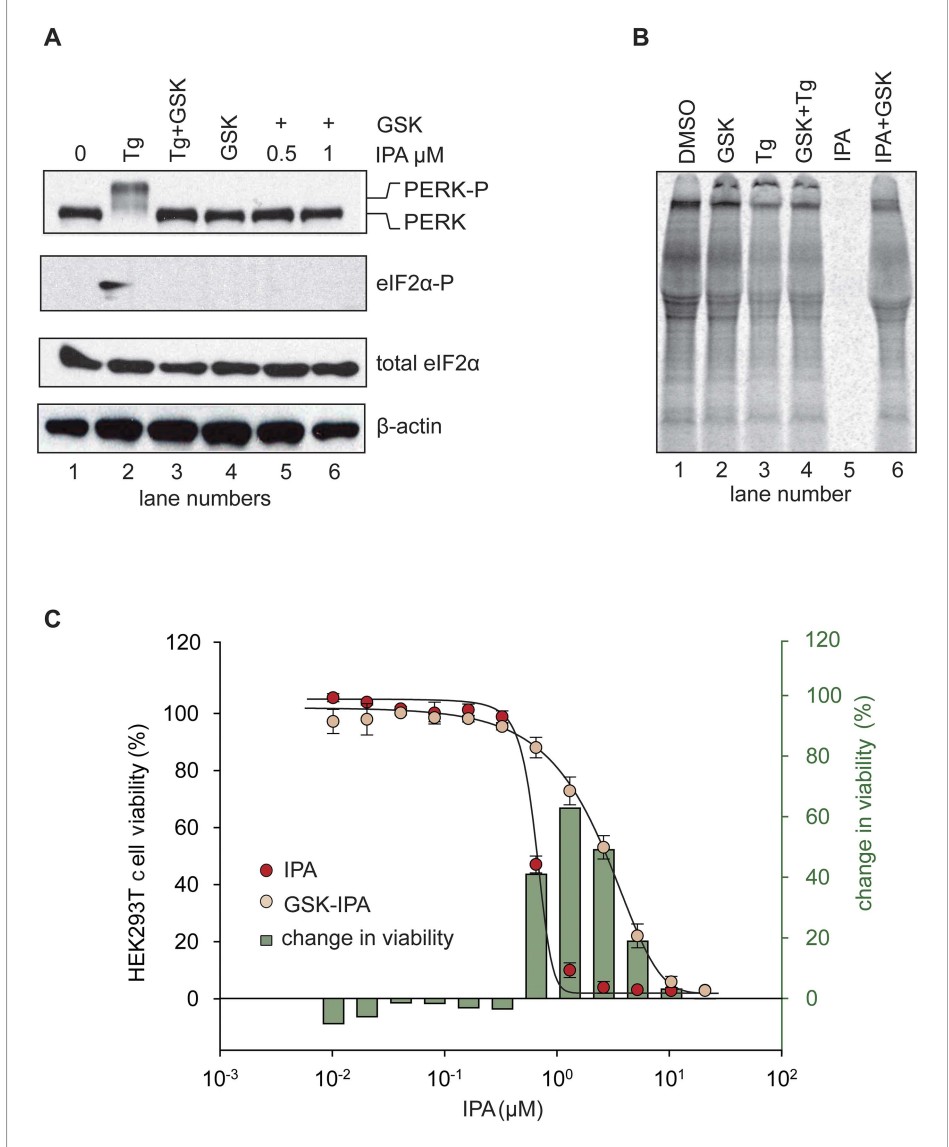

**Figure 5**. Inhibition of IPA-mediated PERK activation by GSK2606414. (**A**) Reversal of IPA-mediated PERK activation ([IPA] = 0.5 and 1 µM; lanes 5 and 6; note peak in *Figure 3C* at these concentrations) in combination with GSK2606414 (GSK, 1 µM) was observed using immunoblotting as described in (*Figure 3C*). As a control, GSK (1 µM) was used to also inhibit PERK branch activation by thapsigargin (Tg; 100 nM; lane 3). (**B**) HEK293T cells were incubated with [35S] methionine to monitor protein translation upon addition of IPA (1 µM) or a combination of IPA (1 µM) and GSK (1 µM). The UPR was induced with Tg (100 nM) or DMSO as indicated. (**C**) Cotreatment of HEK293T cells with IPA + GSK. HEK293T cell viability was measured as a dose-response of IPA in combination with 1 µM of GSK (pink circles). The presence of 1 µM GSK shifted the IPA $LD_{50}$ from 0.82 µM to 6.21 µM. The change in cell viability ± GSK inhibitor is overlaid on both dose responses (green bars). Cells were treated with compounds for 24 hr and cell viability was normalized to DMSO controls.

The following figure supplements are available for figure 5:

**Figure supplement 1**. Quantification of [35S] incorporation.

**Figure supplement 2**. The effects of GSK on HEK293T cell viability.

*Figure 5. Continued*

**Figure supplement 3**. The effect of staurosporine (STS) on HEK293T cell viability in the presence of GSK PERK inhibitor.
**Figure supplement 4**. Inhibition of IPA-mediated PERK activation using GSK2606414.

*Figure 5—figure supplement 2*). Indeed, GSK added at 1 μM protected cells from IPA-induced cell death, shifting the $LD_{50}$ from 0.8 μM to 6.2 μM (*Figure 5C*). As a further control, we tested whether GSK would also rescue cells treated with STS, a rather pleiotropic kinase inhibitor. GSK had no effect under these conditions (*Figure 5C* and *Figure 5—figure supplement 3*).

To address directly whether PERK activation resulted from IPA binding to PERK itself, we recombinantly expressed the cytosolic portion of human PERK and eIF2α, the only known PERK substrate, and reconstituted the phosphorylation reaction in vitro. To this end, the cytosolic portion of PERK was dephosphorylated and then proteolytically severed and chromatographically separated from its glutathione S-transferase tag, known to promote-constitutive multimerization. The resulting PERK cytosolic fragment was responsive to activation, by contrast to the constitutively active, phosphorylated PERK-GST fusion protein assayed in *Figure 4—figure supplement 1*.

As shown in *Figure 6A*, we observed a large increase in the rate of PERK phosphorylation of eIF2α, which peaked at 0.5 μM IPA (3.2-fold increase over DMSO control) and then declined at higher IPA concentrations, whereas IPAx had no effect (*Figure 6A*, *Figure 6—figure supplements 1, 2*). The in vitro bell-shaped activity profile mimicked closely the effect observed in intact cells (*Figure 3C*). Furthermore using chemical crosslinking, we observed that IPA—but not its inactive analog IPAx—significantly increased PERK oligomerization, apparent as an enhanced formation of cross-linked species migrating at the size expected for PERK dimers, trimers, and tetramers (*Figure 6B*). The formation of higher-ordered PERK oligomers during ER stress was previously reported (*Bertolotti et al., 2000*; *Marciniak et al., 2006*).

## Discussion

The kinase domain of IRE1 can be targeted by small molecules to modulate its function. Based on the co-crystal structure of *S. cerevisiae* Ire1 with APY29, we developed a series of new small molecule activators, including IPA. We showed that IPA is a strong activator of IRE1α signaling in vitro, by trapping IRE1α's kinase domain in its active (DFG/αC-in) conformation, which promotes IRE1 oligomerization. In the oligomer, IRE1's RNase becomes activated (*Korennykh et al., 2009*), offering the unique opportunity to read-out the conformational status of a kinase domain in the absence of phospho-transfer. Conversely, a compound, AD60, that traps IRE1's kinase domain in its inactive (DFG/αC-out) conformation acts as an inhibitor of IRE1 signaling. Our studies confirm in mammalian cells that IRE1's kinase domain acts as a conformational switch, in which ligand binding to the ATP binding pocket—rather than enzymatic phospho-transfer—controls activity and down-stream signal transduction events (*Korennykh et al., 2011*).

To rule out that IPA exerts its IRE1 activating activity by causing general ER stress, we explored its effects on the two other branches of the UPR that signal through ATF6 and PERK. Unexpectedly, we discovered that IPA also activated PERK. By contrast to IRE1 however, for which IPA drives activation in vitro and in vivo as IPA concentrations were increased, PERK activation displayed a bell-shaped dose response: PERK was *activated* at low IPA concentrations while being *inhibited* at higher ones.

The paradoxical activation of kinase signaling by kinase inhibitors was first noted for Raf kinase inhibitors almost 15 years ago (*Hall-Jackson et al., 1999*). The observation of kinase activation in patients undergoing Raf inhibitor (vemurafenib) clinical trials then led to an intense investigation of the cellular basis for the phenomenon (*Hatzivassiliou et al., 2010*; *Poulikakos et al., 2010*). Current models suggest that drug-induced dimerization of Raf causes an increase in Raf kinase activity. This contention is supported by apo- and drug-kinase complex crystal structures, which show Raf dimers in the asymmetric crystal unit (*Rajakulendran et al., 2009*). Other models have been proposed that Raf

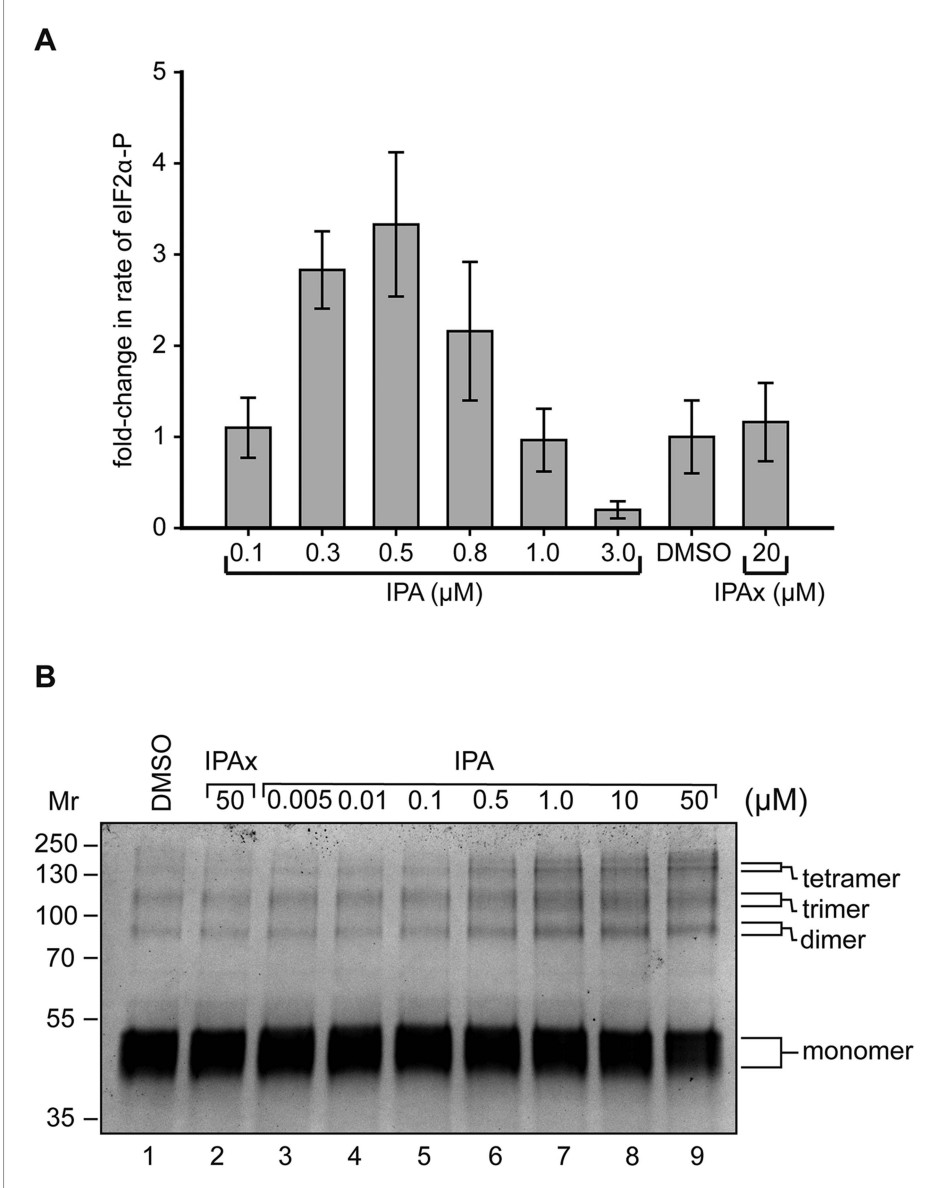

**Figure 6**. Reconstitution of activation of cytosolic PERK protein in vitro. (**A**) Recombinant PERK cytoplasmic domain was incubated at a set concentration of IPA. The fold-change in the rate of eIF2α was normalized to the DMSO control and plotted for all concentration. The greatest effects were observed at 500 nM (3.3-fold change) and 3 µM (0.31-fold change) in activity. IPAx showed no effect on the rate of PERK activity at a concentration of 20 µM. (**B**) Recombinant PERK cytoplasmic domain (2 µM) was preincubated with varying concentrations of IPA (or IPAx) and subjected to chemical cross-linking. An IPA-dependent increase in the dimer, trimer, and tetramer complexes was observed, whereas IPAx (50 µM) showed no effect when compared to the DMSO control.

The following figure supplements are available for figure 6:

**Figure supplement 1**. Biochemical reconstitution of PERK activation.

**Figure supplement 2**. The effect of IPAx on PERK activation in vitro.

inhibitor-induced Raf activation (*Holderfield et al., 2013*), and, to date, the critical prediction of the model that the Raf kinase inhibitor vemurafenib should induce Raf kinase activity in a purified system has resisted all attempts at biochemical reconstitution. This paucity in direct experimental access has

forced mechanistic studies of inhibitor-induced activation to be carried out in cells, where the effects of the many components of the Ras-Raf-Mek-Erk pathway are necessarily confounding. Our work with the reconstitution of PERK's paradoxical activation by IPA shows for the first time that no other components of the kinase pathway are necessary and that inhibitor binding is sufficient for activation through the formation of homo-dimers or higher-ordered homo-oligomers. It was previously suggested that PERK activation is depended upon dimerization and the formation of higher-ordered oligomers (*Bertolotti et al., 2000*).

This behavior can be explained in a model in which ligand binding to a few kinase molecules biases them towards the DFG/αC-in conformation that nucleates assembly with apo-kinases, enabling their *trans*-activation (*Figure 7*) (*Korennykh and Walter, 2012*). Assuming a $K_d$ of 2.8 μM for IPA-binding to PERK as determined by PERK inhibition (*Figure 4A*), we estimate that approximately 15% of the PERK molecules are occupied by IPA under assay conditions that yield maximal activation. Since PERK•IPA triggered activation of unoccupied apo-PERK molecules would bias their conformation towards the active state and hence is likely to enhance their affinity for the inhibitor, the estimate of IPA-occupancy in the population defines an upper limit. This back-of-the-envelope calculation therefore suggests that PERK may form oligomers larger than dimers, in which one PERK•IPA may suffice to activate more than one apo-PERK. At low concentrations of IPA, heterodimers of PERK•IPA/PERK$^{apo}$, or perhaps larger oligomers of PERK•IPA/[PERK$^{apo}$]$_n$, where all PERK molecules are in the active DFG/αC-in conformation, are active in phospho-transfer. At higher, saturating IPA doses, all PERK molecules would be occupied by IPA and thus inhibited through competition with ATP binding.

Thus, the PERK/IPA enzyme/ligand combination mimics the RAF/vemurafenib combination and broadens the number of examples where an ATP-competitive binder can activate rather than inhibit a kinase. This mode of kinase regulation, now documented for two different protein kinases, may be more widespread than currently appreciated. Indeed, we observed a small amount of activation of eIF2α phosphorylation in PERK$^{-/-}$ cells (*Figure 3F*, lane 6), indicating that at least one of the other known eIF2α kinases (GCN2, HRI, and PKR) may be similarly regulated. Kinase activation by partial occupancy of the active site, therefore, may be an important consideration in dosing kinase inhibitors for therapeutic applications, imposing a 'minimal tolerable dose' below which potential drugs may exert detrimental effects that oppose the desired therapy. This dangerous scenario may result from many kinase inhibitors that bind and stabilize kinases in the DFG/αC-in conformation.

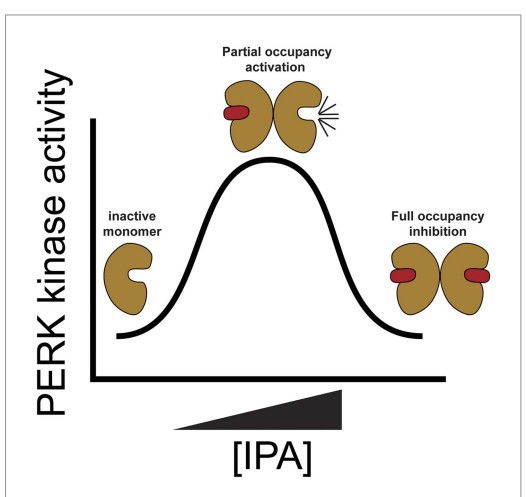

**Figure 7**. Proposed model for PERK activation. The mechanism of PERK activation suggests that at low concentrations of IPA PERK protein is hyper activated presumably through movements of the (DFG/αC-in). At higher doses of IPA all active sites are filled blocking PERK's activity.

Based on the observation that ATF6 was not activated in cells treated with IPA, we conclude that IPA-mediated activation of IRE1 and PERK occurred in the absence of ER stress. This property contrasts with that of the previously described activator 1NM-PP1, which binds in the ATP binding pocket to IRE1-as, the cognate, analog-sensitive allele. Activation of IRE1-as by 1NM-PP1 requires the additional induction of ER stress (*Wang et al., 2012*), or overproduction of IRE1-as driving IRE1-as oligomerization by mass action (*Ghosh et al., 2014*). IPA therefore presents a unique pharmacological tool with which activation of wild-type IRE1 can be studied in living cells directly in the absence of ER stress.

The select application of combinations of UPR modulators has allowed us to begin dissecting the individual contributions of the signaling branches of the UPR. In particular, we have shown that the lethality that IPA displays at high doses can be partially overcome if the PERK pathway is inactivated using a selective PERK inhibitor, GSK, increasing the EC$_{50}$ with which IPA drives cells into apoptosis by about an order of magnitude. This observation is consistent with

the proposed roles of both the IRE1 and PERK branches, providing cytoprotective and pro-apoptotic outputs, respectively (*Lin et al., 2007*). Moreover, AD60, as a DFG-out kinase inhibitor, reversed the effects of IPA on IRE1. IPA, AD60, and other compounds developed to date therefore provide a stepping-stone towards developing novel methodologies for the selective pharmacological tuning of the UPR. Diseases, such as multiple myeloma, a cancer of highly secretory plasma cells where the UPR is thought to play a major cytoprotective role (*Carrasco et al., 2007*; *Leung-Hagesteijn et al., 2013*), or triple-negative breast cancer, in which high XBP1 activity has been correlated with poor patient prognosis (*Chen et al., 2014*), have exposed the potential significance of targeting the UPR in cancers. Additionally, mutations found in cancer cells that weaken IRE1α RNase activity may be amenable targets for allosteric modulation. The chemical biology tools developed in this work provide an important step forward towards exploring the utility of UPR-based therapies, as well as offer fundamental mechanistic insights into a key mechanism that keeps the healthy balance of protein folding in the ER.

## Materials and methods

### Cell culture

HEK293T, T-REx293 and wild type, *Ire1*$^{-/-}$ and *Perk*$^{-/-}$ cells MEF cells were maintained at 37°C, 5% $CO_2$ in Dulbecco's Modified Eagle Medium (DMEM) supplemented with 10% Fetal Bovine Serum (FBS), 10 units/ml penicillin and 10 µg/ml streptomycin (Life Technologies, San Francisco, CA). Tm was obtained from Sigma (Milwaukee, WI). Transient and stable transfections were performed using the Lipofectamine 2000 (Invitrogen, Carlsbad CA) and FuGene6 reagent (Roche). Stable cell lines expressing IRE1-3F6HGFP T-REx293 were described previously (*Li et al., 2010*).

### Live cell imaging

T-REx293 cells were split 2 days before imaging onto glass-bottom micro-well dishes (MatTek) at 5 × $10^4$ cells/dish. Doxycyclin-containing medium (10 nM doxycycline) was added for 24 hr, withdrawn before imaging and replaced with imaging media (Hank's Balanced Salt Solution [Gibco], 2% FBS, and 5 mM HEPES pH 7.0). Images were acquired on a spinning-disk confocal microscope as described (*Li et al., 2010*).

### Metabolic labeling

HEK293T cells (500,000 cells/ml) were plated in 12-well flat bottom culture cluster dishes (Costar, Fisher Scientific) 24 hr before the experiment. IPA (1 µM), or a combination of IPA with GSK2606414 (TRC inc. Toronto Canada), were added to the cells culture for no longer than 2 hr. 15 min prior to lysing cells, 4 µCi of Express [$^{35}$S] protein-labeling mix (Perkin–Elmer) was added. Media were removed, and cells were immediately lysed in a buffer containing 25 mM Tris-HCl pH 8, 8 mM $MgCl_2$, 1 mM dithiothreitol (DTT), 1% Triton X-100, 15% glycerol, 10 mM leupeptin, 153 µM aprotinin, and 1 mM phenylmethanesulfonyl fluoride. Cells lysates were kept on ice for 30 min before high-speed centrifugation followed by denaturation at 95°C. The samples were resolved on an SDS-polyacrylamide gel, which were then stained with Coomassie blue to ascertain protein loading. Dry gels were exposed to a blank phosphorscreen and scanned using a Typhoon variable mode imager (GE Healthcare). The resulting autoradiograms were quantified using ImageQuant software (Molecular Dynamics).

### Cell viability assays

HEK293T cells were grown in DMEM (Sigma) complete media containing 10% FBS, 10 units/ml penicillin (Invitrogen), and 10 µg/ml streptomycin (Invitrogen). Cells were plated at a density of 30,000 cells per well in 96-well black plates with clear flat bottoms (Corning, Sigma-Aldrich) 24 hr before the experiment. A 12-point semi-logarithmic dilution series of compounds was made starting at 30 µM not to exceed 0.1% DMSO after final dilution into the growth media. Viability assays were conducted over the course of 24 hr. Cell viability was determined using CellTiter-Blue (Promega) following the manufacturer's instructions. A Spectramax5 Microplate reader equipped with SoftMax Pro 5 software (Molecular Devices) was used to read out viability, and data were plotted using SigmaPlot software package (Systat Software). Experiments were done in triplicate, and mean values were computed for each data point.

## RNA isolation and RT-PCR

Cells were lysed and total RNA was collected (Total RNA Kit 1, EZ-RNA, EZNA [USA]). PolyA$^+$ mRNA was reverse transcribed with M-MLV (Invitrogen), and the resulting cDNA was used as a template for PCR amplification across the fragment of *XBP1* cDNA containing the intron. Primers used to amplify human *XBP1*: 5′-TTACGAGAGAAAACTCATGGC-3′ and 5′-GGGTCCAAGTTGTCCAGAATGC-3′. To amplify murine *XBP1:* 5′-GAACCAGGAGTTAAGAACACG-3′ and 5′-AGGCAACAGTGTCGAGTCC-3′. PCR conditions were as follows: 95℃ for 5 min, 95℃ for 1 min, 58℃ for 30 s, 72℃ for 30 s, and 72℃ for 5 min, with 20 cycles of amplification. PCR products were resolved on a 2.5% agarose tris-acetate EDTA (TAE) gel. As previously reported, a hybrid amplicon species consisting of unspliced Xbp-1 annealed to spliced Xbp-1 can also be produced through this PCR reaction and was visible as a slower migrating band above the unspliced amplicon. The same cDNA was used as template for PCR of *DERLIN-3* and *HERPUD1*, both ATF6 targets, and *GAPDH*, which was used as loading control. Primers used included the following: *DERLIN-3*, 5′-AGTTCCACTCTTTGATGGAGGGCA-3′ and 5′-AGCCAGCTGTGAGGAATATGGGAA-3′; *HERPUD1-1*, 5′-ACAAGGTGGCCCTATTGTGGAAGA-3′ and 5′-AGTCCATTCCTGTCAAAGCCTCCA-3′; *GAPDH*, 5′-CCATGTTCGTCATGGGTGTGA-3′ and 5′-CATGGACTGTGGTCATGAGT-3′. PCR conditions were: 95℃ × 5 min, 95℃ × 1 min, 56℃ × 30 s, 72℃ for 30 s for 25 cycles, 72℃ for 5 min at 4℃. PCR products were resolved on a 2% agarose TAE gel.

## Protein expression and purification

Human IRE1α-KR43 with a hexa-histidine (6xHis) tag in its N-terminus was expressed and purified in SF21 cells as described (*Li et al., 2010*). Cells were lysed in buffer containing 20 mM Tris-HCl, pH 7.5, 600 mM NaCl, 2 mM MgCl$_2$, 3 mM imidazole, 10% glycerol, 1% Triton X-100, 3 mM β-mercaptoethanol, COMPlete protease inhibitors (Roche), and PhosSTOP phosphatase inhibitor cocktail (Roche) and passed through an AVESTIN emulsiflex-C3 3× times for complete lysis. The lysate was cleared using centrifugation at 100,000×*g*. Clear lysate was allowed to incubate on Ni-NTA agarose (Qiagen) beads for 2 hr before being washed 3× times with buffer containing 20 mM Tris-HCl, pH7.5, 600 mM NaCl, 2 mM MgCl$_2$, 30 mM imidazole, 10% glycerol, and 3 mM β-mercaptoethanol. Protein was eluted from the column by raising the imidazole concentration to 250 mM. The eluate was then passed through a HiTrap desalting column (GE Healthcare). Lastly, the 6xHis was cleaved as described (*Li et al., 2010*) and the resulting protein was then loaded onto a HisTrap HP, 5 × 5 ml column to remove the uncleaved protein. The cleaved protein was then loaded on a Mono-S 5/50 GL column (GE Healthcare), and the eluate was then concentrated to 5 mg/ml and loaded on a Superdex 200 HR 10/300 (GE Healthare) column in buffer containing 20 mM Tris-HCl pH 7.5, 250 mM NaCl, 5% Glycerol and 5 mM tris(2-carboxyethyl) phosphine (TCEP). The pure protein solution was aliquoted and stored at −80℃.

To express the cytosolic kinase domain of PERK, the murine PERK kinase domain (580–1077aa) was cloned into a pGEX4T1 vector to create a fusion protein containing N-terminal GST. The plasmid was transformed into *Escherichia coli* strain BL21DE3 RIPL (Agilent Technologies). Cells were grown in LB medium containing ampicillin and chloramphenicol until OD600 = 0.6. Expression was induced with 0.2 mM IPTG at 18℃ for 16 hr. Cells were harvested by centrifugation, resuspended in buffer A (50 mM Tris pH 7.5, 150 mM NaCl, 5 mM MgCl$_2$, 3 mM β-mercaptoethanol, 2.5% Glycerol) and lysed by sonication. After centrifugation, the supernatant was applied to a GST-Sepharose column and washed with buffer A and buffer A containing 500 mM KCl and 1 mM ATP. Sample was eluted with buffer A containing 20 mM glutathione. GST-PERK kinase domain was then concentrated and further purified on a Superdex 200 10/300 gel filtration column equilibrated in buffer B (50 mM Tris pH 7.5, 50 mM NaCl, 5 mM MgCl$_2$, 3 mM DTT, 1% Glycerol). Fractions containing GST-PERK kinase domain were concentrated and flash-frozen in liquid nitrogen and stored at −80℃.

Full-length cytosolic human PERK was codon-optimized for *E. coli* expression by Genewiz Inc. A construct was then cloned into a Pgex-6p-2 vector for expression using two rounds of In-Fusion cloning (Clontech) (535–1093 Δ660–868). The cytosolic portion of PERK, lacking the unstructured loop region (amino acids 535–1093 Δ660–868) was then co-expressed with a tagless lambda phosphatase to produce a fully dephosphorylated PERK protein in BL21 star (DE3) (Life Technologies) (*Dar et al., 2008*). Cells were grown to an OD$_{600}$ of 0.5 before induction with 0.1 mM IPTG at 15℃ for 25 hr. Cells were harvested and lysed using AVESTIN Emulsiflex-C3 in a buffer containing 50 mM Tris-HCl, pH 8.0, 500 mM NaCl, 10% glycerol, 5 mM TCEP (buffer A), and EDTA-free COMPlete protease inhibitor cocktail (Roche). Lysate was cleared by centrifugation at 100,000×*g* before batch binding to a GST-Sepharose

resin. The resin was washed 5× times with buffer A, and on-column tag cleavage was preformed using PreScission protease (GE Healthcare). The protein was loaded onto a HiTrap Q HP column to remove remaining protease. The PERK (535–1093 Δ660–868) protein was then concentrated and fractionated on a Superdex 200 GL (GE Healthcare) to remove uncleaved GST-PERK protein.

## Immunoblotting

Cells were lysed in 50 mM Tris-HCl, 2% SDS, 10% glycerol, 0.01% bromophenol blue, pH 6.8 supplemented with phosphatase inhibitors (Sigma) and protease inhibitors (Roche). 30 µg to 50 µg of total protein was loaded on each lane of 8% or 10% SDS-PAGE gels. The separated proteins were then transferred onto nitrocellulose membranes for immunoblotting. The following antibodies and dilutions were used: anti-β-actin 1:10,000 (A5441, Sigma), anti-total PERK at 1:1000 (C33E10, Cell Signaling Technology), anti-phospho eIF2α at 1:1000 (3597S, Cell Signaling Technology), anti-total eIF2α at 1:1000 (L57A5, Cell Signaling Technology). After overnight incubation with primary antibody, membranes were washed in PBS with 0.05% Tween and incubated in HRP-coupled secondary antibody anti-rabbit (611–1302, Rockland) or anti-mouse (610–1302, Rockland) diluted at 1:5000 in wash buffer. Immunoreactive bands were detected by chemiluminescence.

## In vitro endoribonuclease IRE1 activity assay

IRE1 endoribonuclease activity was detected employing two in vitro assays. $k_{obs}$ and Hill coefficients were determined from the cleavage kinetics of [$^{32}$P]-labeled RNA substrates as previously described (*Korennykh et al., 2009*). The assay was started by adding 1 µl of [$^{32}$P]-labeled RNA to 9 µl of premixture containing 20 mM HEPES pH 7.4, 70 mM NaCl, 2 mM MgCl$_2$, 4 mM DTT, 5% glycerol, 1 µl of 10 µM compound in DMSO. Reactions were performed at 30°C. Reactions were conducted under single turnover conditions. They contained ≤1 pM [$^{32}$P]-labeled RNA and increasing concentrations of purified IRE1α-KR43. Reactions were quenched at time intervals with 6 µl stop solution (10 M urea, 0.1% SDS, 0.1 mM EDTA, 0.05% xylene cyanol, and 0.05% bromophenol blue). Samples were analyzed in 10–20% Urea-PAGE gels. Gels were scanned using a Typhoon variable mode imager (GE Healthcare) and quantified using the ImageQuant and GelQuant software packages (Molecular Devices). The data were plotted and fit to exponential curves using SigmaPlot software package (Systat Software) to determine observed rate constants as previously described (*Korennykh et al., 2009*). EC$_{50}$ values were determined using a FRET-based cleavage assay. A FRET probe (FRET_IDT_17 56- FAM/rCrArCrCrUrCrUrGrCrArGrCrArGrGrUrG/ IABlk_FQ) was purchased from IDT (RNase free HPLC purification) and dissolved in RNase-free H$_2$O. Excitation: 485 (480–490) and emission: 517 (512–520). The reactions were started by addition of RNA FRET probe (to a final concentration of 100 nM) to 9.4 µl of premixture containing 20 mM HEPES, pH 7.4, 70 mM NaCl, 2 mM MgCl$_2$, 4 mM DTT, 5% glycerol, and compound in DMSO not to exceed 1%. Reactions were performed at 30°C and contained 150 nM enzyme under single turnover conditions. Reactions were quenched at time intervals with equal volumes of formamide. Samples were analyzed using a SpectraMax M5 plate reader equipped with SoftMax Pro 5 and data acquisition software (Molecular Devices). The data were plotted using SigmaPlot software package (Systat Software). Experiments were repeated 3–4 times and mean values were computed.

## Generation of XBP1 reporter cell line

A plasmid encoding a modified ER stress reporter construct consisting of a C-terminally FLAG-tagged firefly luciferase coding sequence fused to an N-terminal hemagglutinin-tagged tandem repeat of a partial sequence of XBP1 of human origin lacking its DNA-binding domain and containing the IRE1-cognate intron (pCAX-HA-2xXBP1ΔDBD(anATG)-LUC-F, kind gift of Takao Iwawaki [*Hosoda et al., 2010*]), was used as a template to generate a retroviral expression construct. The coding sequence of the reporter was amplified by PCR using primers with engineered BamHI and EcoRI sites and was subsequently cloned into the cognate sites of the retroviral expression vector pBABE.puro (Addgene) to generate construct DAA-A171. DAA-A171 was used to generate recombinant retroviral particles using standard methods and the resulting retroviral supernatant was used to transduce HEK293T cells, which were then subsequently selected with puromycin to create a stable reporter cell line.

## Kinase-inhibitor profiling

IPA and **cmp1** were assayed by Invitrogen to derive percent inhibition of kinase activity. All compounds were screened at 1 µM and raw values are shown in supplementary tables. Detailed

procedures for kinase reactions, ATP concentrations used and Z′-LYTE or Adapta assay formats are described in SelectScreen Customer Protocol (http://www.invitrogen.com/kinaseprofiling). Kinase-inhibition data for 1 µM inhibitor of each of the clinical and tool compounds STS, Sunitinib, Dasatinib, Imatinib, SB202190, Erlotinib and Gefitinib were provided by Invitrogen based on stock results and not an independent comparison conducted or commissioned by the authors. Raw data of percent inhibition of all 266 kinases can be located in *Supplementary file 1*. Screening analysis for APY24 was previously reported (*Statsuk et al., 2008*).

## In vitro PERK cross-linking assay

A solution of PERK cytosolic domain (535–1093 Δ660–868) (2 µM) was incubated with IPA or IPAx for 15 min before the addition of 250 µM of the amino group-reactive cross-linking agent bis [sulfosuccinimidyl] suberate (BS2g-d0, Thermo Scientific). After 30-min incubation at room temperature Tris-HCl pH 7.5 was added to a final concentration of 55 mM. Samples were then boiled in Laemmli sample buffer and loaded into an Any kD Mini-protean TGX gel (Bio Rad). Cross-linked protein was visualized using Colloidal Coomassie G-250 stain.

## In vitro PERK kinase assay

GST-PERK kinase domain (580–1077 aa) was pre-incubated with compounds for 30 min. The kinase reaction was initiated by addition of ATP including 0.2 mCi γ-[$^{32}$P]-ATP. The final concentrations of the reactants were 50 nM GST-PERK, 50 µM eIF2α, 100 µM ATP and varying concentrations of compound. At 30 min, 2 µl of each reaction was spotted onto a P81 phospho-cellulose membrane. The membrane was washed in 1% phosphoric acid and the sheets were washed five times in buffer, dried, and transferred radioactivity was measured using a Typhoon variable mode imager (GE Healthcare) and quantified using the ImageQuant (Molecular Devices). Titration data were fit to a sigmoidal dose response to derive IC$_{50}$ values using the SigmaPlot software package (Systat software). Dose responses were based on a 12-point inhibitor titration, using a semi-logarithmic dilution series starting from 30 µM. Experiments were completed 2–4 times and mean values were computed.

PERK activation was measured using PERK (535–1093 Δ660–868) in a buffer containing 20 mM Tris-HCl, 150 mM NaCl, 4 mM MgCl$_2$, 5 mM TCEP, 1% glycerol, pH 8.0. PERK concentration was 1 µM, and activity was measured at 30°C. PERK was pre-incubated with 50 µM of cold ATP in the presence or absence of IPA or IPAx. The reaction was then initiated by adding *S. cerevisiae* eIF2α (1–180) and 0.2 mCi γ-[$^{32}$P]-ATP. The reaction was monitored over a time course of 30 min, during which 1/6 of the reaction volume was removed at 0.5, 1, 5, 10, 20, and 30 min and stopped in Laemmli buffer supplemented with 50 mM EDTA. Reactions were then boiled and loaded onto a 12.5% Criterion precast gel (Bio-rad). The gel was then dried before imaging using a Typhoon variable mode imager (GE Healthcare). Gels were quantified using ImageQuant software and data were processed using SigmaPlot software package. Rates were determined using non-linear regression. Experiments were performed 3 times, and mean errors were determined.

The fraction PERK bound by IPA was calculated using the standard Langmuir isotherm equation. Using the calculated IC$_{50}$ described in *Figure 4A*.

$$\text{fraction bound} = \frac{1}{1 + \frac{IC_{50}}{[I]}}.$$

## Chemistry: synthesis of intermediate 3

2,4-dichloropyrimidine **2** (13.30 g, 89.31 mmol) and 3-amino-5-cyclopropylpyrazole **1** (11.00 g, 89.31 mmol) were dissolved in tetrahydrofuran (TFA) (100 ml) followed by the addition of deionized water (100 ml). The solution was then treated with potassium acetate (261.9 g, 2.5 mol). The resulting mixture was kept at 55°C for 48 hr. The mixture was filtered to remove excess potassium acetate. The organic layer was separated and dried using magnesium sulfate and concentrated before loading on an AnaLogix silica column. Intermediate **3** was separated as previously reported (*Hosoda et al., 2010*). [1]H NMR (400 MHz, DMSO) δ 12.14 (s, 1H), 10.23 (s, 1H), 1.84 (m, 1H), 0.88 (m, 2H), 0.64 (m, 2H). [13]C NMR (100 MHz, DMSO) δ 161.4, 160.7, 160.0, 153.9, 148.6, 147.8, 146.8, 8.4, 8.2. MS calculated for $C_{10}H_{10}N_5$ 235.06, found 236.15 (M[+]).

## Synthesis of intermediate 5

Pyrimidine monochloride **3** (4 g, 16.9 mmol) (Oakwood chemical) and p-phenylene-diamine **4** (1.9 g, 16.9 mmol) (Sigma) were dissolved in butanol (BuOH) (50 ml) (Sigma) followed by the addition of concentrated HCl (0.1 ml) (Fisher Scientific). The resulting mixture was kept at 100°C overnight. Purple precipitate was collected by filtration, washed with 30 ml of cold BuOH and dried under vacuum yielding intermediate **5**, which was used in the next step without further purification. MS calculated for $C_{16}H_{17}N_7$ 307.15, found 308.5.

## Synthesis of cmp1

Intermediate **5** (153 mg, 0.5 mmol) was dissolved in 10 ml of dry dimethylformamide (DMF) (Sigma). The flask was placed in an ice bath before addition of 3-(trifluoromethyl)phenyl isocyanate (93.5 mg; 0.069 ml; 0.5 mmol) **6** (Sigma) drop wise using a Hamilton syringe under argon. The resulting mixture was allowed to warm to room temperature overnight. Crude compound was isolated by the addition of cold $H_2O$ (30 ml) and isolated by filtration. Solid was washed with 100% $CH_3CN$ and then suspended in 1:1 $CH_3CN$ and distilled water ($dH_2O$) for reverse-phase purification using HPLC. A gradient from 2–80% $CH_3CN$:$H_2O$ (0.1% TFA) was used to further during purification on a C18 column (Agilent Technologies) followed by lyophilization to yielded **cmp1**. [1]H NMR (400 MHz, DMSO-d$_6$): δ 12.30 (s, 1H), 10.00 (s, 2H), 8.70 (s, 3H), 7.43 (m, J = 2.3 Hz, 6H), 6.62 (d, J = 2.1 Hz, 5H), 5.29 (s, 1H) 2.51 (m, J = 10.2 Hz, 4H), 1.79 (t, J = 9.6 Hz, 1H), 0.87 (s, 4H) [13]C NMR (151 MHz, DMSO-d$_6$) δ 155.7,

152.8, 141.8, 138.4, 135.5, 133.6, 130.2, 129.3, 128.9, 118.8, 112.6, 97.6, 8.9; ESI-MS m/z [M + H] found 495.49 calculated 494.49.

## Synthesis of cmp2

Intermediate **5** (153 mg, 0.5 mmol) was dissolved in 10 ml of dry DMF. Flask was placed in an Ice bath before addition of p-Toyl isocyanate (0.066 mg; 0.063 ml; 0.5 mmol) **7** (Sigma) drop wise using a Hamilton syringe under argon. Resulting mixture was allowed to warm to room temperature overnight. Crude compound was isolated by addition of cold $H_2O$ (30 ml) and isolated by filtration. Solid was washed with 100% $CH_3CN$ and then suspended in 1:1 $CH_3CN$ and $dH_2O$ for reverse-phase purification using HPLC. A gradient from 2–80% $CH_3CN:H_2O$ (0.1% TFA) was used to further during purification on a C18 column (Agilent Technologies) followed by lyophilization to yield **cmp2**. [1]H NMR (400 MHz, DMSO-d$_6$): δ 12.21 (s, 1H), 9.13 (s, 1H), 8.58 (q, J = 1.3 Hz, 9H), 8.35 (d, J = 21.6 Hz, 3H), 6.54 (s, 3H) 2.51 (m, J = 10.2 Hz, 4H), 1.89 (s, 1H) 1.29 (t, J= 1.1 Hz, 3H), 0.87 (s, 4H); [13]C NMR (151 MHz DMSO-d$_6$) δ 155.4, 153.8, 146.8, 140.3, 139.7, 136.7, 131.2, 127.8, 117.3, 115.4, 63.5, 33.5, 32.1 8.1; ESI-MS m/z [M + H] found 440.28 calculated 440.21.

## Synthesis of cmp3

Intermediate **5** (153 mg, 0.5 mmol) was dissolved in 10 ml of dry DMF. The flask was placed in an ice bath before addition of 4-Ethylphenyl isocyante (0.073 mg; 0.072 ml; 0.5 mmol) **8** (Sigma) drop wise using a Hamilton syringe under argon. The resulting mixture was allowed to warm to room temperature overnight. Crude compound was isolated by addition of cold $H_2O$ (30 ml) and isolated by filtration. Solid was washed with 100% $CH_3CN$ and then suspended in 1:1 $CH_3CN$ and $dH_2O$ for reverse-phase purification using HPLC. A gradient from 2–80% $CH_3CN:H_2O$ (0.1% TFA) was used to further during purification on a C18 column (Agilent Technologies) followed by lyophilization to yield product **cmp3**. [1]H NMR (400 MHz, DMSO-d$_6$): δ 12.30 (s, 1H), 10.00 (s, 2H), 7.50 (d, J = 2.1 Hz, 3H), 7.33 (s, 6H), 7.08 (t, J = 7.2 Hz, 3H), 1.79 (s, 2H), 1.13 (m, J = 13.4 Hz, 5H), 0.87 (s, 4H); [13]C NMR (151

MHz DMSO-d$_6$) δ 162.4, 153.0, 141.16, 130.4, 130.0, 129.9, 123.8, 122.21, 118.4, 114.5, 113.9, 35.6, 8.4; ESI-MS m/z [M + H] found 455.5 calculated 454.31.

## Synthesis of cmp4

Intermediate **5** (153 mg, 0.5 mmol) was dissolved in 10 ml of dry DMF. The flask was placed in an ice bath before addition of 3-methoxyphenyl isocyanate (0.075 mg; 0.066 ml; 0.5 mmol) **9** (Sigma) drop wise using a Hamilton syringe under argon. The resulting mixture was allowed to warm to room temperature overnight. Crude compound was isolated by the addition of cold H$_2$O (30 ml) and isolated by filtration. Solid was washed with 100% CH$_3$CN and then suspended in 1:1 CH$_3$CN and dH$_2$O for reverse-phase purification using HPLC. A gradient from 2–80% CH$_3$CN:H$_2$O (0.1% TFA) was used to further during purification on a C18 column (Agilent Technologies) followed by lyophilization to yield **cmp4**. $^1$H NMR (400 MHz, DMSO-d$_6$): δ 12.43 (s, 1H), 11.1 (s, 1H), 10.17 (s, 1H), 8.78 (d, $J$ = 19 Hz, 2H), 8.68, 7.82 (s, 1H), 7.57 (q, $j$ = 6.1 Hz, 2H), 7.41 (t, $J$ = 8.9 Hz, 2H), 7.18 (q, $J$ = 7.2 Hz, 3H), 6.84 (d, J = 5.8 Hz, 1H), 3.89 (s, 3H), 1.79 (s, 1H), 0.87 (s, 4H); $^{13}$C NMR (151 MHz DMSO-d$_6$) δ 160.1, 152.95, 141.4, 130.0, 119.9, 110.9, 107.6, 104.4, 55.3, 8.37; ESI-MS m/z [M + H] found 457.6 calculated 456.45.

## Synthesis of cmp5

Intermediate **5** (153 mg, 0.5 mmol) was dissolved in 10 ml of dry DMF. The flask was placed in an ice bath before addition of 4-methoxyphenyl isocyanate (0.075 mg; 0.065; 0.5 mmol) **10** (Sigma) drop wise using a Hamilton syringe under argon. The resulting mixture was allowed to warm to room temperature overnight. Crude compound was isolated by the addition of cold H$_2$O (30 ml) and isolated by filtration. Solid was washed with 100% CH$_3$CN and then suspended in 1:1 CH$_3$CN and dH$_2$O for reverse-phase purification using HPLC. A gradient from 2–80% CH$_3$CN:H$_2$O (0.1% TFA) was used to further during purification on a C18 column (Agilent Technologies) followed by lyophilization to yield **cmp5**. $^1$H NMR (400 MHz, DMSO-d$_6$): δ 12.40 (s, 1H), 11.18 (s, 1H), 10.20 (s, 1H), 9.73 (q, $J$ = 8.6 Hz, 2H), 8.78 (d, $J$ = 8.9 Hz, 3H), 7.62 (t, $J$ = 2.5 Hz, 2H), 7.44 (t, $j$ = 6.1 Hz, 3H), 6.98 (t, $J$ = 7.2 Hz,

2H) 3.78 (m, $J = 12.2$ Hz, 3H), 1.79 (s, 1H), 0.87 (s, 4H), 0.54 (s, 2H); $^{13}$C NMR (151 MHz DMSO-d$_6$) δ 159.3, 153.1, 141.3, 125.3, 121.4, 110.4, 108.6, 104.3, 58.3, 8.4; ESI-MS m/z [M + H] found 457.5 calculated 456.51.

## Synthesis of cmp6 (IPA)

Intermediate **5** (153 mg, 0.5 mmol) was dissolved in 10 ml of dry DMF. The flask was placed in an ice bath before addition of 4-(methylthio)phenyl isocyanate (0.082 mg; 0.07 ml; 0.5 mmol) **11** (Sigma) drop wise using a Hamilton syringe under argon. The resulting mixture was allowed to warm to room temperature overnight. Crude compound was isolated by addition of cold H$_2$O (30 ml) and isolated by filtration. Solid was washed with 100% CH$_3$CN and then suspended in 1:1 CH$_3$CN and dH$_2$O for reverse-phase purification using HPLC. A gradient from 2–80% CH$_3$CN:H$_2$O (0.1% TFA) was used to further during purification on a C18 column (Agilent Technologies) followed by lyophilization to yield **IPA**. $^1$H NMR (400 MHz, DMSO-d$_6$): δ 12.10 (s, 1H), 9.95 (s, 2H), 8.61 (d, $J = 3.7$ Hz, 2H), 7.48 (d, $J = 2.3$ Hz, 8H), 7.22 (d, $J = 21.3$ Hz, 3H), 2.21 (m, $J = 10.2$ Hz, 3H), 1.79 (s, 1H), 0.87 (s, 4H); $^{13}$C NMR (151 MHz DMSO-d$_6$) δ 161.3, 155.3, 142.3, 140.6, 137.4, 135.8, 133.3, 131.2, 128.3, 119.5, 114.3, 16.5, 8.2; ESI-MS m/z [M + H] found 472.18 calculated 472.20.

## Synthesis of cmp7

Intermediate **5** (153 mg, 0.5 mmol) was dissolved in 10 ml of dry DMF. The flask was placed in an ice bath before addition of 4-iodophenyl isocyanate (0.122 mg, 0.5 mmol) **12** (Sigma) drop wise using a Hamilton syringe under argon. The resulting mixture was allowed to warm to room temperature overnight. Crude compound was isolated by addition of cold H$_2$O (30 ml) and isolated by filtration. Solid was washed with 100% CH$_3$CN and then suspended in 1:1 CH$_3$CN and dH$_2$O for reverse-phase purification using HPLC. A gradient from 2–80% CH$_3$CN:H$_2$O (0.1% TFA) was used to further during purification on a C18 column (Agilent Technologies) followed by lyophilization to yield **cmp7**. $^1$H NMR (400 MHz, DMSO-d$_6$): δ 12.21 (s, 1H), 9.10 (s, 1H), 8.81 (s, 1H), 8.79 (s, 1H), 8.52 (t, $J = 8.5$ Hz, 1H), 8.08

(S, 1H), 7.68 (m, J = 11.8 Hz, 4H), 7.38 (m, $J$ = 10.2, 4H), 6.14 (d, $J$ = 15.2 Hz, 2H), 1.79 (s, 1H), 0.87 (s, 4H); $^{13}$C NMR (151 MHz DMSO-d$_6$) δ 156.4, 152.9, 140.3, 137.7, 136.1, 133.5, 120.7, 120.2, 119.3, 116.9, 98.1, 84.7, 8.1; ESI-MS m/z [M + H] found 553.3 calculated 552.08.

## Synthesis of cmp8

Intermediate **5** (153 mg, 0.5 mmol) was dissolved in 10 ml of dry DMF. The flask was placed in an ice bath before addition of 4-bromophenyl isocyanate (0.099 mg, 0.5 mmol) **13** (Sigma) drop wise using a Hamilton syringe under argon. The resulting mixture was allowed to warm to room temperature overnight. Crude compound was isolated by addition of cold H$_2$O (30 ml) and isolated by filtration. Solid was washed with 100% CH$_3$CN and then suspended in 1:1 CH$_3$CN and dH$_2$O for reverse-phase purification using HPLC. A gradient from 2–80% CH$_3$CN:H$_2$O (0.1% TFA) was used to further during purification on a C18 column (Agilent Technologies) followed by lyophilization to yield **cmp8**. $^1$H NMR (400 MHz, DMSO-d$_6$): δ 12.21 (s, 1H), 9.51 (s, 1H), 8.81 (m, $J$ = 21.8 HZ, 2H), 8.68 (d, $J$ = 6.3 Hz, 1H), 7.82 (s, 1H), 7.48 (m, $J$ = 22.5 Hz, 8H), 7.33 (d, $J$ = 23.2 Hz, 2H), 7.08 (t, $J$ = 7.2, 3H), 1.79 (s, 1H), 0.87 (s, 4H); $^{13}$C NMR (151 MHz DMSO-d$_6$) δ 155.3, 152.3, 140.9, 138.3, 137.6, 132.6, 122.4, 121.8, 120.8, 118.4, 95.4, 82.7, 8.2; ESI-MS m/z [M + H] found 505.5 calculated 504.10.

## Synthesis of cmp9

Intermediate **5** (153 mg, 0.5 mmol) was dissolved in 10 ml of dry DMF. The flask was placed in an ice bath before addition of 4-isopropylphenyl isocyanate (0.080 mg; 0.08; 0.5 mmol) **14** (Sigma) drop wise using a Hamilton syringe under argon. The resulting mixture was allowed to warm to room temperature overnight. Crude compound was isolated by addition of cold H$_2$O (30 ml) and isolated by filtration. Solid was washed with 100% CH$_3$CN and then suspended in 1:1 CH$_3$CN and dH$_2$O for reverse-phase purification using HPLC. A gradient from 2–80% CH$_3$CN:H$_2$O (0.1% TFA) was used to further during purification on a C18 column (Agilent Technologies) followed by lyophilization to yield **cmp9**. $^1$H NMR (400 MHz, DMSO-d$_6$): δ 12.11 (s, 1H), 8.51 (t, $J$ = 13.6 Hz, 4H), 7.58 (m, $J$ = 8.3 Hz, 2H),

7.22 (q, $J$ = 12.6 Hz, 8H), 6.48 (d, $J$ = 5.1 Hz, 1H), 1.13 (m, $J$ = 13.4, 8H), 0.87 (m, $J$ = 19.4 Hz, 4H); $^{13}$C NMR (151 MHz DMSO-d$_6$) δ 158.3, 151.6, 144.7, 139.3, 138.6, 136.9, 134.5, 124.8, 119.1, 118.5, 36.5, 32.5, 8.1; ESI-MS m/z [M + H] found 469.5 calculated 468.24.

## Synthesis of cmp10

Intermediate **5** (200 mg, 0.5 mmol) was dissolved in 10 ml of dry DMF. The flask was placed in an ice bath before addition of 4-tert-butylphenyl isocyanate (0.087 mg; 0.089 ml; 0.5 mmol) **15** (Sigma) drop wise using a Hamilton syringe under argon. The resulting mixture was allowed to warm to room temperature overnight. Crude compound was isolated by addition of cold H$_2$O (30 ml) and isolated by filtration. Solid was washed with 100% CH$_3$CN and then suspended in 1:1 CH$_3$CN and dH$_2$O for reverse-phase purification using HPLC. A gradient from 2–80% CH$_3$CN:H$_2$O (0.1% TFA) was used to further during purification on a C18 column (Agilent Technologies) followed by lyophilization to yield **cmp10**. $^1$H NMR (400 MHz, DMSO-d$_6$): δ 12.11 (s, 1H), 9.43 (s, 1H), 8.81 (s, 1H), 8.68 (m, $J$ = 22.3 Hz, 2H), 7.92 (s, 1H), 7.43 (m, $j$ = 28.1 Hz, 8H), 6.28 (t, $J$ = 7.2 Hz, 2H) 1.79 (s, 1H), 1.13 (s, 9H), 0.87 (s, 4H); $^{13}$C NMR (151 MHz DMSO-d$_6$) δ 153.1, 144.3, 137.7, 135.8, 133.9, 125.8, 119.1, 118.3, 119.07, 118.3, 34.3, 31.7, 30.9, 8.1; ESI-MS m/z [M + H] found 483.5 calculated 482.25.

## Synthesis of IPAx

Pyrimidine monochloride **3** (0.4 g, 1.67 mmol) was dissolved in dry round bottom flask containing 30 ml of dichloromethane (anhydrous) (Sigma). The flask was then chilled to 4°C in an ice bath under argon. Potassium carbonate (K$_2$CO$_3$) (0.92 g, 6.68 mmol) was added, and the reaction was stirred for 10 min. Iodomethane (0.425 g, 2.5 mmol) was then added drop-wise and kept at 4°C until determined complete by thin layer chromatography. The reaction was then filtered to remove excess K$_2$CO$_3$ and an equal volume of dH$_2$O was added to reaction mixture. The mixture was washed three times with brine solution and then dried using magnesium sulfate. The solution was then filtered and concentrated before being loaded on to an AnaLogix silica column. Monomethylated intermediate **16** was isolated using a solution of 5% methanol in chloroform over 30 min. ESI-MS m/z [M + H] found 250.3 calculated 249.09.

Intermediate **16** (0.157 g, 0.633 mmol) and p-phenylene-diamine (0.068 g, 0.633 mmol) were dissolved in BuOH (3 ml) followed by the addition of concentrated HCl (0.01 ml) resulting mixture was kept at 100°C for 8 hr. Isolation and purification of **17** is as described for intermediate **3**. ESI-MS m/z [M + H]$^+$ found 322.5 calculated 321.19.

Intermediate **17** (0.038 mg, 0.123 mmol) was dissolved in 5 ml of dry DMF. The flask was placed in a ice bath before addition of 4-(methylthio)phenyl isocyonate (0.023 mg, 0.123 mmol) (Sigma) drop wise using a Hamilton syringe under argon. The resulting mixture was allowed to warm to room temperature overnight. Crude compound was isolated by the addition of cold $H_2O$ (30 ml) and isolated by filtration. Solid was washed with 100% $CH_3CN$ and then suspended in 1:1 $CH_3CN$ and d$H_2O$ for reverse-phase purification using HPLC. A gradient from 2–80% $CH_3CN$:$H_2O$ (0.1% TFA) was used to further during purification on a C18 column (Agilent Technologies) followed by lyophilization to yield **IPAX**. $^1$H NMR (400 MHz, DMSO-d$_6$): δ 10.00 (s, 2H), 8.81 (s, 1H), 8.68 (d, J = 2.3 Hz, 1H), 7.82 (s, 1H), 7.50 (d, j = 2.1 Hz, 3H), 7.33 (s, 6H), 7.08 (t, J = 7.2, 3H), 6.08 (s, 2H), 1.79 (s, 1H), 0.87 (s, 4H); $^{13}$C NMR (151 MHz DMSO-d$_6$) δ 162.4, 153.0, 141.1, 131.3, 130.1, 129.3, 123.8, 122.1, 118.2, 115.2, 114.2, 8.4; ESI-MS m/z [M + H]$^+$ found 487.5 calculated 486.21.

## Acknowledgements

We thank the members of the Walter and Shokat labs for helpful advice and support. We also acknowledge Drs Erica L Cain, Jirka Peschek, Shelly Starck, and Chao Zhang for guidance in assay development and many invaluable insights. This work was funded by a Collaborative Innovation Award from the Howard Hughes Medical Institute to SB, KMS and PW and by PFB16/Fondecyt 1131137 from the Comisión Nacional de Investigación Científica y Tecnológica, Chile, to SB, JA and MAMS. KMS and PW are Investigators of the Howard Hughes Medical Institute.

## Additional information

### Competing interests

ACD, An inventor on University of California, San Francisco patent (Patent Application Number: US20120322814 A1). HL, An inventor on University of California, San Francisco patent (Patent

Application Number: US20120322814 A1). PW, An inventor on University of California, San Francisco patent (Patent Application Number: US20120322814 A1). AVK, An inventor on University of California, San Francisco patent (Patent Application Number: US20120322814 A1). KMS, An inventor on University of California, San Francisco patent (Patent Application Number: US20120322814 A1). The other authors declare that no competing interests exist.

## Funding

| Funder | Grant reference | Author |
|---|---|---|
| Howard Hughes Medical Institute (HHMI) | Collaborative Innovation Award | Sebastian Bernales, Kevan M Shokat, Peter Walter |
| Howard Hughes Medical Institute (HHMI) | Investigators | Kevan M Shokat, Peter Walter |

The funder had no role in study design, data collection and interpretation, or the decision to submit the work for publication.

## Author contributions

ASM, Conceived and designed experiments with SB, KMS and PW. ASM and ACD performed chemical synthesis and characterization, ASM performed in vitro biochemical measurements and cell drug combinations, ASM, KMS and PW wrote the manuscript with input from all authors, Conception and design, Acquisition of data, Analysis and interpretation of data, Drafting or revising the article; JA, MAM-S, Conceived and designed experiments with SB, KMS and PW, Performed in cell characterization of IPA, Conception and design, Acquisition of data, Analysis and interpretation of data, Drafting or revising the article; ACD, Performed chemical synthesis and characterization, Conception and design, Drafting or revising the article; EMC, Analysis and interpretation of data, Drafting or revising the article; KG, Provided PERK-GST fusion protein and eIF2α, Drafting or revising the article, Contributed unpublished essential data or reagents; HL, Performed imaging of IRE1 foci, Acquisition of data, Drafting or revising the article; DA-A, CS, Generated the XBP1 reporter cell line, Drafting or revising the article, Contributed unpublished essential data or reagents; AVK, Helped with analysis, Analysis and interpretation of data, Drafting or revising the article; SB, Conceived and designed experiments, Conception and design; KMS, PW, Conceived and designed experiments, Conception and design, Drafting or revising the article

# Additional files

### Supplementary file

• Supplementary file 1. Invitrogen Kinome screen raw data. All kinases screened in the Invitrogen panel and their percent inhibitions. Classification of kinases can be located on the right hand column and the number of kinases inhibited can be located at the bottom of the last page of the panel.

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
