## [Decision Letter]

Thank you for sending your work entitled “ER stress-independent activation of unfolded protein response kinases by a small molecule ATP-mimic” for consideration at *eLife*. Your article has been favorably evaluated by John Kuriyan (Senior editor and Reviewing editor) and three reviewers.

The Reviewing editor and the reviewers discussed their comments before we reached this decision, and the Reviewing editor has assembled the following comments to help you prepare a revised submission.

In this study, Mendez et al. report the discovery and characterization of IPA, a small molecule (ATP mimetic) modulator of the unfolded protein response effectors IRE1 and PERK.

Previous work by the authors using small molecule ATP-mimics to inhibit the kinase domain of IRE1 showed an interesting mechanism by which IRE1 responds to unfolded protein stress: The kinase activity of the IRE1 kinase is not required, but rather the active conformation of the kinase.

In the present work, using biochemical assays, the authors demonstrate that IPA is an activator of IRE1 both in vitro and in vivo. Based on previous experiments published by the authors, the data are interpreted with a model in which the new compound shifts the equilibrium between the active and inactive conformation of the kinase much more towards the active state.

Unexpectedly, they also report that IPA acts as a PERK activator at low concentrations and as a PERK inhibitor at high concentrations. The authors propose that at low (non-saturating) concentrations of IPA, IPA binding to one PERK protomer promotes dimerization and transactivation of a second PERK protomer bound to ATP, resulting in auto-phosphorylation of PERK and subsequent phosphorylation of eIF2α. At higher concentrations of IPA, all PERK protomers are occupied by IPA resulting in complete inhibition of its phosphotransfer function.

Experiments in *IRE1*^*-/-*^ MEFs show that activation of PERK does not involve IRE1, and in vitro experiments confirm direct binding of the compound to PERK. Treatment of cells with GSK2606414, a potent PERK inhibitor, provided a protective effect against IPA toxicity, providing further support for the model that IPA toxicity is mediated by off-target inhibition of PERK.

This is an interesting and timely observation as it closely mimics the analogous phenomenon uncovered for RAF-specific kinase inhibitors, which cause paradoxical ligand-induced activation of RAF kinase signaling in cells. It thus appears that the notion of paradoxical activation of kinase function by kinase inhibitors may be more general to kinase families regulated by kinase domain dimerization. These observations have important implications for the design and clinical use of protein kinase small-molecule therapeutics.

Overall the manuscript is well written, easy to follow and the data are of high quality. Aspects of the work dealing with structure-informed design of the compound, its specificity and in vitro and cell-based activity on IRE1 are clear and convincing. Despite the general interest in the work, the reviewers all feel that the conclusions regarding the model for PERK activation must be strengthened, with particular attention to the following concerns/questions, described below under the “Major Points” heading. A collated set of other comments that should be considered is also provided.

Major points:

1) While the data are consistent with the proposed model for PERK activation, there are no data to directly show this. There are quite straightforward biochemical experiments that could deliver such evidence:

i) Given that IPA strongly stabilizes the active conformation of IRE1, one would expect quite strong cooperative binding of IPA to PERK if the proposed model is correct: Binding of IPA to PERK causes the neighboring apo-PERK molecules to be in the active state, hence the high affinity state. This seems to be a good experiment to validate the proposed model.

ii) Since the authors have the cytosolic part of PERK expressed, a more quantitative in vitro characterization of the binding of the compound would strengthen the proposed model.

2) In Figure 4, the authors use recombinant GST-PERK to demonstrate that IPA directly inhibits PERK. Specifically they show that IPA inhibits PERK phosphorylation of eIF2α in vitro. However, no evidence of enhancement of kinase activity is observed at low concentrations of IPA that mimics the phenomenon observed in cells (i.e. IPA activates PERK kinase function in cells at low IPA concentration) as shown in Figure 3.

In contrast, at the very end of the manuscript, the authors point to Figure 6 showing that low concentrations of IPA in fact does activate PERK kinase activity against eIF2α in vitro. What is the basis for this discrepancy? Perhaps both data panels could be shown in the same figure for clarity.

3) In Figure 5, the authors show that at low concentrations of IPA, PERK is activated leading to impaired translation (lane 5 vs. lane 3). However, they fail to show what happens to translation at high concentrations of IPA. If their model is correct, then at higher concentrations of IPA, cellular translation should be restored and match the levels shown in lanes 3, 4 and 6. The presented figure in its current form is so striking and tantalizing that the reader hungers for a more detailed analysis.

Other points to consider:

Reviewer 1:

1) At the end of the subsection headed “In vivo effects of IPA on the ATF6 and PERK UPR branches”, the authors state that IPA is an ATP competitive inhibitor. This is not proven experimentally so the authors should either provide the data in support (ATP competition curves) or qualify the original statement (the compound is likely an ATP competitive inhibitor based on the following facts…).

2) The authors should also include the autoradiographs quantified in Figure 4 as a supplementary figure.

3) In Figure 5, Mendez et al. show that PERK signaling can be turned off using GSK (a previously published high affinity PERK selective inhibitor developed by GSK) even in the presence of low concentrations of IPA, that alone cause activation of PERK. One essential missing control in this data figure is the IPA-alone control lanes. To be consistent with other figures presented in the manuscript, the authors should include 1 micromolar (PERK activation) and 4 micromolar (PERK inhibition) concentrations of IPA.

4) In Figure 5, the authors should include a negative control showing the growth inhibition profile of GSK alone, even if it doesn't inhibit growth.

5) In the fourth paragraph of the Discussion section, how do the authors infer the K_d_ from the IC^50^? Include the details in the Methods section.

6) In Figure 6, the authors perform a full dose response analysis for IPA but only show a single point analysis for IPAx. And the concentration tested for IPAx does not match any of the concentration points tested for IPA. As such, the IPAx analysis is not informative. The authors should redo the experiment using comparable concentrations of IPAx.

7) Figures 1 and 2 are consistent with the notion that IPA induces the dimerization of Ire1. In the RAF field, authors use reciprocal IPs to demonstrate inhibitor-induced dimerization in cells (and AUC in vitro). Can the authors provide direct evidence that IPA induces PERK dimerization using either immunofluorescence (as per Figure 2) or IP pull downs? This data would strengthen the proposed model.

Reviewer 2:

1) Is IPA less toxic to *PERK*^*-/-*^ MEFs than WT or *IRE1*^*-/-*^ MEFs?

2) Need controls in Figure 5 with IPA treatment but no GSK. Also the thapsigargin induced shift in “phosphoPERK” mobility looks qualitatively more pronounced than that induced by IPA. Is it possible that there are different sites getting phosphoryalted?

3) Why aren't higher concentrations of IPA (sufficient to inhibit PERK) non-toxic?

4) Discussion regarding IPA-induced dimerization of PERK is over emphasized, given that there are no data in the paper examining effect of IPA on oligomeric state of PERK. Fine to point out recent findings in RAF, but would not include Figure 7 without experimentally addressing the question.

5) It would be helpful to remind the reader that AD60 is expected to bind an inactive conformation of IRE1, and therefore not to activate RNase (in the appropriate place in the Results section).

6) Define IRE1 (KR43) construct in Results.

7) Make clear in the text that specificity comparisons with APY24 and the various commercial TKIs are based on “stock” results from invitrogen, and not an independent comparison conducted/commissioned by the authors (if this is indeed the case?)

8) Does the GSK compound bind an active or inactive PERK kinase conformation?

Reviewer 3:

1) It is very difficult to see that the IRE1α-GFP foci become less but larger with Tm treatment, and stay small but numerous with IPA. Could the authors improve the figure? Second what is the mechanistic explanation for the different behavior, particularly the growth of the foci with Tm?

2) The authors talk about “oligomerization” that triggers RNase activity, can they be more specific about what oligomeric state is found?

3) The finding for PERK activation is in analogy to ATP-competitive inhibitor induced RAF dimerization. Since the dimerization mechanism is the heart of the manuscript (Figure 7), dimerization as a function of IAP concentration needs to be shown with additional direct experiments as previously done for RAF using AUC.

---

## [Author Response]

*Major points*:

*1) While the data are consistent with the proposed model for PERK activation, there are no data to directly show this. There are quite straightforward biochemical experiments that could deliver such evidence*:

*i) Given that IPA strongly stabilizes the active conformation of IRE1, one would expect quite strong cooperative binding of IPA to PERK if the proposed model is correct: Binding of IPA to PERK causes the neighboring apo-PERK molecules to be in the active state, hence the high affinity state. This seems to be a good experiment to validate the proposed model*.

*ii) Since the authors have the cytosolic part of PERK expressed, a more quantitative in vitro characterization of the binding of the compound would strengthen the proposed model*.

We have added additional data to address this point (Figure 6). To determine whether IPA oligomerizes PERK in vitro we performed a chemical cross-linking experiment where we titrated IPA at a constant concentration of PERK cytoplasmic domain. We observed clear enhancement of discrete higher molecular weight species at increased IPA concentrations, which demonstrate IPA-driven enhanced PERK dimer, trimer, and tetramer formation. By contrast, the control compound IPAx did not enhance oligomer formation over the DMSO-only control.

*2) In*
Figure 4*, the authors use recombinant GST-PERK to demonstrate that IPA directly inhibits PERK. Specifically they show that IPA inhibits PERK phosphorylation of eIF2α in vitro. However, no evidence of enhancement of kinase activity is observed at low concentrations of IPA that mimics the phenomenon observed in cells (i.e. IPA activates PERK kinase function in cells at low IPA concentration) as shown in*
Figure 3.

*In contrast, at the very end of the manuscript, the authors point to*
Figure 6
*showing that low concentrations of IPA in fact does activate PERK kinase activity against eIF2α in vitro. What is the basis for this discrepancy? Perhaps both data panels could be shown in the same figure for clarity*.

We have clarified the point in the text and added compound titration data as Figure 4—figure supplement 1. In brief, the murine PERK-GST fusion protein is the best-behaved construct we had generated. Due to its propensity to dimerize mediated by the GST moiety, recombinantly produced PERK-GST fusion protein is constitutively phosphorylated and enzymatically active. Thus using this construct, we could not capture the activation stage of PERK. To show the activation effects of IPA, we expressed after considerable effort a fully dephosporylated tagless construct.

*3) In*
Figure 5*, the authors show that at low concentrations of IPA, PERK is activated leading to impaired translation (lane 5 vs. lane 3). However, they fail to show what happens to translation at high concentrations of IPA. If their model is correct, then at higher concentrations of IPA, cellular translation should be restored and match the levels shown in lanes 3, 4 and 6. The presented figure in its current form is so striking and tantalizing that the reader hungers for a more detailed analysis*.

At higher IPA concentrations translation remains inhibited. This is likely due to off-target effects, perhaps elicited by activating another eIF2α kinase(s). We have added a statement to this effect to the manuscript.

*Other points to consider*:

Reviewer 1:

*1) At the end of the subsection headed “In vivo effects of IPA on the ATF6 and PERK UPR branches”, the authors state that IPA is an ATP competitive inhibitor. This is not proven experimentally so the authors should either provide the data in support (ATP competition curves) or qualify the original statement (the compound is likely an ATP competitive inhibitor based on the following facts…)*.

The appropriate change was made to the manuscript in the Discussion section.

*2) The authors should also include the autoradiographs quantified in*
Figure 4
*as a supplementary figure.*

We included the autoradiographs in Figure 4—figure supplement 1. Also see our response to Major Point 2) above.

*3) In*
Figure 5*, Mendez et al. show that PERK signaling can be turned off using GSK (a previously published high affinity PERK selective inhibitor developed by GSK) even in the presence of low concentrations of IPA, that alone cause activation of PERK. One essential missing control in this data figure is the IPA-alone control lanes. To be consistent with other figures presented in the manuscript, the authors should include 1 micromolar (PERK activation) and 4 micromolar (PERK inhibition) concentrations of IPA.*

We have attached two controls showing activation of PERK at 0.5 and 1 µM along with PERK inhibition with the GSK inhibitor to Figure 5—figure supplement 4.

*4) In*
Figure 5*, the authors should include a negative control showing the growth inhibition profile of GSK alone, even if it doesn't inhibit growth*.

We show in Figure 5—figure supplement 2 that cell viability in HEK293T cells is unaffected by the GSK compound at the concentrations used in our experiments.

*5) In the fourth paragraph of the Discussion section, how do the authors infer the K*_*d*_
*from the IC*_*50*_*? Include the details in the Methods section*.

The methodology and equation we used can be found in the Methods section of the manuscript, in the subsection headed “In vitro PERK kinase assay”. In brief, we used the standard Langmuir-isotherm to determine the fraction bound. We made the assumption that binding of IPA to the GST-PERK construct was unaffected by phosphorylation state and the constitutive dimer induced by GST. We then used the IC_50_ values derived from this experiment (Figure 4) along with the inhibitor concentration that we observed the most robust activation to determine a back-of-the-envelope estimate of occupancy. Further mechanistic analyses of occupancy will be performed in more biophysically focused follow-up studies that however are beyond the current scope of this work.

*6) In*
Figure 6*, the authors perform a full dose response analysis for IPA but only show a single point analysis for IPAx. And the concentration tested for IPAx does not match any of the concentration points tested for IPA. As such, the IPAx analysis is not informative. The authors should redo the experiment using comparable concentrations of IPAx*.

We have redone the experiment using two concentrations where we see the greatest effect on PERK activation in vitro (0.3 and 0.5 µM) and included the data in Figure 6—figure supplement 2. As a positive control we also used 0.5 µM of IPA to show the activation profile. As expected, we observed no effect in PERK activation with IPAx when normalized to DMSO.

*7)*
Figures 1 and 2
*are consistent with the notion that IPA induces the dimerization of Ire1. In the RAF field, authors use reciprocal IPs to demonstrate inhibitor-induced dimerization in cells (and AUC in vitro). Can the authors provide direct evidence that IPA induces PERK dimerization using either immunofluorescence (as per*
Figure 2*) or IP pull downs? This data would strengthen the proposed model*.

We have provided new data (Figure 6) showing that IPA induces PERK oligomerization through chemical cross-linking (see response to Major Point 1 above) to provide concrete evidence of IPA-driven PERK oligomerization.

Author response image 1.**DOI:**
http://dx.doi.org/10.7554/eLife.05434.023

*Reviewer 2*:

*1) Is IPA less toxic to* PERK^-/-^
*MEFs than WT or* IRE1^-/-^
*MEFs?*

IPA is less toxic in *PERK*^*-/-*^ cells (IC_50_ 9.1 µM) when compared to WT MEF cells (IC_50_ 4.3 µM). We included the necessary graph for reviewing purposes that illustrates this point.

*2) Need controls in*
Figure 5
*with IPA treatment but no GSK. Also the thapsigargin induced shift in “phosphoPERK” mobility looks qualitatively more pronounced than that induced by IPA. Is it possible that there are different sites getting phosphoryalted?*

We have attached a new figure with the appropriate controls in Figure 5—figure supplement 4. The point raised by the reviewer concerns treatment time and differences in the UPR inducer (Tm vs. Tg). In the earlier figures with IPA treatment cells were incubated for 4 hours. In the case of Figure 5 the treatment was also done for 4 hours but Tm-treated cells are slower in inducing a UPR response, whereas Tg cells elicit the response faster and more robustly.

3) Why aren't higher concentrations of IPA (sufficient to inhibit PERK) non-toxic?

See response to Major Point 3 above.

*4) Discussion regarding IPA-induced dimerization of PERK is over emphasized, given that there are no data in the paper examining effect of IPA on oligomeric state of PERK. Fine to point out recent findings in RAF, but would not include*
Figure 7
*without experimentally addressing the question.*

We have added concrete evidence of PERK oligomerization (see response to Major Point 1 above).

5) It would be helpful to remind the reader that AD60 is expected to bind an inactive conformation of IRE1, and therefore not to activate RNase (in the appropriate place in the Results section).

We placed the additional description of AD60 in the text of the Results section.

6) Define IRE1 (KR43) construct in Results.

We defined the construct in the Results section with the exact length of the IRE1α linker. Further details are located in the Methods section.

7) Make clear in the text that specificity comparisons with APY24 and the various commercial TKIs are based on “stock” results from invitrogen, and not an independent comparison conducted/commissioned by the authors (if this is indeed the case?)

We made the appropriate changes to the text in the Methods section.

8) Does the GSK compound bind an active or inactive PERK kinase conformation?

It is thought to bind to the inactive and active conformer of PERK (1).

Reviewer 3:

1) It is very difficult to see that the IRE1α-GFP foci become less but larger with Tm treatment, and stay small but numerous with IPA. Could the authors improve the figure? Second what is the mechanistic explanation for the different behavior, particularly the growth of the foci with Tm?

We improved the figure. This difference in behavior results from the attenuation of IRE1α foci under normal ER-stress (23). IPA blocks this step, activating IRE1α constitutively.

2) The authors talk about “oligomerization” that triggers RNase activity, can they be more specific about what oligomeric state is found?

Based off our hill analysis IRE1α can form a minimum of 4.2 molecules in an active oligomer with IPA. The method in which the hill coefficient is calculated for IRE1 has been published prior (19). We do understand that while this suggest a tetramer there could be bigger oligomers coming together as seen in live imaging.